# Aromatic acids in a Eurasian Arctic ice core: a 2,600-year proxy record of biomass burning

Mackenzie M. Grieman[1], Murat Aydin[1], Diedrich Fritzsche[2], Joseph R. McConnell[3], Thomas Opel[2,4], Michael Sigl[5], and Eric S. Saltzman[1]

[1]Department of Earth System Science, University of California, Irvine, Irvine, California, 92697-3100, USA
[2]Alfred Wegener Institute Helmholtz Centre for Polar and Marine Research, Potsdam, Germany
[3]Division of Hydrologic Sciences, Desert Research Institute, Reno, Nevada, USA
[4]Permafrost Laboratory, Department of Geography, University of Sussex, Brighton, UK
[5]Laboratory of Environmental Chemistry, Paul Scherrer Institut, Villigen, Switzerland

*Correspondence to:* Mackenzie M. Grieman (mgrieman@uci.edu)

**Abstract.** Wildfires and their emissions have significant impacts on ecosystems, climate, atmospheric chemistry and carbon cycling. Well-dated proxy records are needed to study the long-term climatic controls on biomass burning and the associated climate feedbacks. There is a particular lack of information about long-term biomass burning variations in Siberia, the largest forested area in the Northern Hemisphere. In this study we report analyses of aromatic acids (vanillic and para-hydroxybenzoic acids) over the past 2,600 years in the Eurasian Arctic Akademii Nauk ice core. These compounds are aerosol-borne, semi-volatile organic compounds derived from lignin combustion. The analyses were made using ion chromatography with electrospray mass spectrometric detection. The levels of these aromatic acids ranged from below the detection limit (.01 to .05 ppb; 1 ppb = 1,000 ng/l) to about 1 ppb, with roughly 30% of the samples above the detection limit. In the preindustrial late Holocene, highly elevated aromatic acid levels are observed during three distinct periods (650-300 BCE, 340-660 CE, and 1460-1660 CE). The timing of the two most recent periods coincides with the episodic pulsing of ice-rafted debris in the North Atlantic known as Bond events and a weakened Asian monsoon, suggesting a link between fires and large-scale climate variability on millennial time scales. Aromatic acid levels also are elevated during the onset of the industrial period from 1780-1860 CE, but with a different ratio of vanillic and para-hydroxybenzoic acid than is observed during the preindustrial period. This study provides the first millennial scale record of aromatic acids. This study clearly demonstrates that coherent aromatic acid signals are recorded in polar ice cores that can be used as proxies for past trends in biomass burning.

## 1 Introduction

Fire is a major disturbance to net primary production in boreal forest ecosystems (Gower et al., 2001). Determining how fire regimes and ecosystems have changed in the past provides insight into how climate change may influence fire and its impact on the carbon cycle in the future. Terrestrial and lacustrine sedimentary charcoal records are the main source of information about regional variations in past biomass burning. A synthesis of high latitude Northern Hemisphere charcoal records indicates a gradual decline in burning related to Late Holocene cooling, followed by an increase from 1750-1870 CE and a decline after

1870 CE associated with anthropogenic activity (Marlon et al., 2008). Biomass burning increased during the first half of the $20^{th}$ century, declined during the second half of the $20^{th}$ century, and rose sharply after 2000 CE (Marlon et al., 2016). Siberia is the largest forested area in the Northern hemisphere and Siberian wildfires constituted 5 to 20% of global biomass burning carbon emissions from 1998-2002 CE (Soja et al., 2004). There are only 11 Siberian sedimentary charcoal records in the Global Charcoal database (Blarquez et al., 2014). These records cover very different age ranges at varied temporal resolutions. As a result, it is not yet possible to reconstruct Siberian biomass burning trends on centennial or millennial time scales with confidence (Marlon et al., 2008, 2016; Power et al., 2008).

Biomass burning emissions histories also have been inferred from a variety of different ice-core proxies (Legrand et al., 2016; Rubino et al., 2015). Variations in the stable isotopic composition of ice core methane have been used as a proxy for global biomass burning emissions (Ferretti et al., 2005; Sapart et al., 2012). Biomass burning is not the primary source of atmospheric methane as more methane is emitted from geologic and various microbial sources. The contribution from burning is calculated from measurements of the stable isotopic ratio ($^{13}C/^{12}C$) of methane by assigning end-member isotopic compositions to the various sources. Late Holocene methane isotopic records show that global burning emissions were high from 1-1000 CE, declined from 1000-1700 CE, and increased again after 1700 CE (Ferretti et al., 2005; Mischler et al., 2009).

Several other chemicals with shorter atmospheric lifetimes have been used as regional, rather than global, fire proxies. Ammonium has been commonly used as a tracer for biomass burning in several ice cores (Legrand et al., 1992, 2016; Rubino et al., 2015). Elevated levels of ammonium with the same timing as elevated levels of formate, acetate, oxalate, glycolate, formaldehyde, hydrogen peroxide, potassium, nitrate, or black carbon have been interpreted as indications of elevated burning periods in ice cores (Fuhrer et al., 1993, 1996; Legrand et al., 1992; Legrand and De Angelis, 1996; Rubino et al., 2015; Savarino and Legrand, 1998; Taylor et al., 1996; Whitlow et al., 1994; Yalcin et al., 2006). The concurrent timing of ammonium spikes with decreases in electrical conductivity has also been used as an indication of biomass burning in ice cores (Chýlek et al., 1995; Rubino et al., 2015; Taylor et al., 1996). The challenge of using ammonium as a biomass burning tracer is that it has several other sources, including animal excreta, synthetic fertilizers, oceanic sources, crops, natural vegetation soils, lightning, industrial processes, fossil fuels, and other anthropogenic sources (Bouwman et al., 1997; Rubino et al., 2015). Simultaneous timing of ammonium and formate peaks in ice cores have also been used as a proxy for increases in biogenic emissions due to periods of increased temperature (Eichler et al., 2009; Rubino et al., 2015). Ice core proxies that are uniquely derived from burning are needed to confirm the interpretation of ammonium as a biomass burning tracer.

Black carbon in ice cores has been used as a tracer for preindustrial biomass burning (Chýlek et al., 1995; Legrand et al., 2016; McConnell et al., 2007; Rubino et al., 2015; Zennaro et al., 2014). Ammonium, formate, black carbon, and organic carbon (dissolved organic carbon or total organic carbon) were enriched substantially relative to background levels during fire events in Greenland ice (Legrand et al., 2016). During industrial times, black carbon in ice cores also originates from fossil fuel combustion. Differences between black carbon and other biomass burning proxy records in ice cores has been attributed to transport and combustion conditions. Black carbon primarily is produced under flaming combustion conditions, while ammonia is generated by smoldering fires (Rubino et al., 2015). Boreal fires, which are geographically closer to high latitude ice core sites, are often smoldering fires (Legrand et al., 2016).

Wildfires generate a wide range of aerosol-borne organic compounds that are derived from the partial combustion of plant matter. Levoglucosan is an aerosol-borne anhydrous sugar exclusively produced by burning of cellulose (Simoneit et al., 1999). Levoglucosan is a promising tracer because it makes up a large fraction of the organic aerosol mass produced by biomass burning, is emitted from burning of all types of cellulose-containing plant matter, and has been detected in aerosols and ice in polar regions (Simoneit et al., 1999; Kehrwald et al., 2012; Gambaro et al., 2008; Kawamura et al., 2012; Yao et al., 2013; Zennaro et al., 2014). However, the utility of this compound as a quantitative tracer is somewhat controversial due to the potential for rapid degradation in the atmosphere (Hoffmann et al., 2010; Hennigan et al., 2010; Slade and Knopf, 2013).

Wildfires also generate phenolic breakdown products derived from the pyrolysis of lignin during the smoldering stage of a fire (Akagi et al., 2011; Legrand et al., 2016; Simoneit, 2002). The chemistry of these emissions reflects the composition of the precursor lignin and the rate, temperature, and oxidative conditions under which burning occurs. Aromatic acids, such as vanillic acid (VA), para-hydroxybenzoic acid (p-HBA), and syringic acid are molecules that are diagnostic of biomass burning because they retain the basic aromatic building block of the precursor lignin (Fig. S1; Hedges and Mann, 1979; Hedges and Parker, 1976; Opsahl and Benner, 1995; Oros and Simoneit, 2001a, b; Simoneit, 2002; Vanholme et al., 2010). Laboratory burning studies show a range in the yield of different aromatic acids from natural biomass fuels. For example, burning of North American conifers produces both VA and p-HBA, with VA in greater abundance, while North American tundra grass fires produce p-HBA with essentially no VA (Nolte et al., 2001; Oros and Simoneit, 2001a, b; Oros et al., 2006; Otto et al., 2006; Rogge et al., 1998; Simoneit et al., 1993; Simoneit, 2002). German peat fires have been shown to produce both VA and p-HBA, with p-HBA in greater abundance (Iinuma et al., 2007). Surprisingly, there are no published studies examining the burning products of Siberian flora.

The abundance of aromatic acids, such as VA and p-HBA, in polar ice cores reflect the combined effects of biomass burning emissions, atmospheric transport and transformations, depositional processes, and post-depositional processes. VA and p-HBA are semi-volatiles that may reside either in the gas or condensed phase depending on temperature, aerosol water content, pH, and cation concentrations. There is some debate regarding the atmospheric lifetimes of these semi-volatile compounds because in the gas phase they can react rapidly with hydroxyl radicals. The OH lifetime for gas phase oxidation of these compounds is on the order of a day. However, modelling suggests that such compounds are shielded from oxidation inside aerosol particles, with atmospheric lifetimes of several days (Donahue et al., 2013). Such lifetimes are supported by observations of long-distance atmospheric transport of biomass burning aerosols. There are numerous observations of aromatic acids in burning-derived coarse and fine mode atmospheric aerosols in terrestrial, marine, Arctic, and Antarctic environments (Simoneit and Elias, 2000; Simoneit et al., 2004; Zangrando et al., 2013, 2016). The field observations and model estimates of reactivity support the idea that long distance transport is the likely source of these compounds to the ice sheet. Near surface postdepositional processes such as revolatilization, photochemical reactivity, or melt-water infiltration have not been studied for aromatic acids. Such processes could potentially influence the ice core levels of these compounds, particularly at low accumulation sites (Grannas et al., 2007).

FLEXPART model forward trajectories suggest that in the summer aerosol transport to the Arctic from biomass burning sources is primarily from Siberia (48°-66°N, 60°-140°E; Stohl, 2006). The FLEXPART model is a Lagrangian transport and

dispersion model that is used to simulate long-range atmospheric transport. Twenty-five percent of FLEXPART modelled forward trajectories from Siberia reached the Arctic in 3 days, and 50% reached the Arctic in 10 days (Stohl, 2006). Russian fires in 2003 contributed 40-56% of the mass of BC deposited in the spring and summer above 75°N (Generoso et al., 2007). Lidar data from the Arctic Research of the Composition of the Troposphere from Aircraft and Satellites (ARCTAS) mission indicate that biomass burning plumes from Russian forest fires in 2008 contributed to aerosol loadings over the North American Arctic (Matsui et al., 2011; Warneke et al., 2010).

The deposition of burning-derived aerosols to the polar ice sheets and ice caps raises the possibility that ice cores contain well-dated biomass burning records that integrate fire emissions over wide geographic regions. These records are complementary to other historical records of biomass burning, such as sedimentary charcoal and ice core gas records. Although more complex in terms of integrating both emissions and transport, the ice cores bring some advantages to the study of past biomass burning. They are regionally integrated archives of climate and burning proxies that are stored in the same well-dated record. The information stored in aromatic acid records is distinctly different from that contained in ice core gas records of methane stable isotopes, carbon monoxide, or ethane, which are influenced primarily by tropical, rather than boreal or high latitude emissions (Ferretti et al., 2005; Nicewonger et al., 2016; Sapart et al., 2012; Wang et al., 2010).

The distribution of these aerosols in polar ice cores has not been systematically investigated. VA and p-HBA were measured in a 300-year ice core from the Kamchatka Peninsula, Northeast Asia (Kawamura et al., 2012), and VA was measured in a 200-year ice core from the D4 site in west-central Greenland (McConnell et al., 2007). VA, p-HBA, and several other organic biomass burning tracers were measured in an ice core covering part of the $20^{th}$ century from the Swiss Alps (Müller-Tautges et al., 2016).

In this study, VA and p-HBA were measured in an ice core from the Eurasian Arctic (80°31'N, 94°49'E). The 724 m long ice core was drilled on the Akademii Nauk ice cap (5,575 km$^2$, 800 m above sea level) on the Severnaya Zemlya archipelago (Fig.1; Fritzsche et al., 2002, 2005; Opel et al., 2009, 2013; Spolaor et al., 2016; Weiler et al., 2005). This is the longest ice core record of these aromatic acids measured to date, with samples ranging in age from 650 BCE-1988 CE.

## 2 Methods

### 2.1 Ice core sample collection and dating

For the upper 129 m of the Akademii Nauk ice core, the analyses were made on discrete samples that were previously analysed for major ions (Weiler et al., 2005). These samples were melted at the Alfred Wegener Institute from a 3x3 cm-thick cross section of the core, at a resolution of roughly two samples per year. Samples from 129-671 m were melted at the Desert Research Institute by the continuous melting of a 3.2x3.2 cm cross section of the core (after McConnell et al., 2002). Subsamples from the melt stream were collected for this study via fraction collector at a resolution of roughly one sample per year. Samples were stored frozen in polyethylene vials prior to analysis.

The original chronology for the upper 411 m of the core (900-1998 CE) was developed based on annual layer counting of stable water isotopes, a 1963 CE cesium peak, and 5 volcanic sulphate signals tied to Greenland ice cores (Opel et al., 2013). In

this study, we used a new chronology based on continuous flow chemistry. For the upper 407 m (743-1998 CE), we interpolated linearly between a pollution signal in 1955 CE and 9 volcanic sulphur signals (Table S1; Fig. S2; Arienzo et al., 2016; Sigl et al., 2013, 2015). The dating of these tie points was based on correlation with other Arctic ice cores (Sigl et al., 2013, 2015). The original and new age scales are very similar from the surface to the depth of the Laki eruption in 1783 CE at 105 m. Below

this depth, the new age scale yields progressively older ages due to the availability of volcanic tie points at 1330, 1477, 1594, and 743 CE linked to Greenland ice cores (Sigl et al., 2013, 2015). The depth-age scale below 407 m is not constrained by tie points. This part of the age scale was based on linear interpolation of annual layer thickness between the deepest tie point and the bottom of the ice core (0.21 and 0.12 m water equivalent year$^{-1}$, respectively). Layer thickness near the bed was based on oxygen isotope ($\delta^{18}$O), deuterium ($\delta$D), deuterium excess ($\delta$D-8$\delta^{18}$O), and electrical conductivity (DEP) measurements

(Fritzsche et al., 2005). The lowermost portion of this age scale below the tie points is considered provisional. Further analysis of the chemistry of the lowermost portion of the ice core or other dating approaches may result in improvement in the age scale.

### 2.1.1   Analytical methods

The conventional method for analysis of aromatic acids in environmental samples involves pre-concentration, derivatization,

and gas chromatography with mass spectrometric detection (Kawamura et al., 2012; Oros and Simoneit, 2001a; Simoneit et al., 1993). This method requires large sample volumes and complex pre-treatment procedures. In previous work, we reported direct detection of vanillic acid in polar ice via continuous-flow electrospray ionization with tandem mass spectrometry (ESI/MS/MS) (McConnell et al., 2007). That approach provided very high temporal resolution but because it did not involve chromatographic separation, it was potentially subject to positive interferences from other organics present in the ice core samples. Grieman et al.

(2015) developed a method for analysing aromatic organic acids in discrete ice core samples using HPLC-ESI/MS/MS, with chromatography providing additional selectivity. HPLC-ESI/MS/MS has also been used to measure aromatic acids in Arctic and Antarctic atmospheric aerosol samples (Zangrando et al., 2013, 2016).

For the current study, analysis of VA, p-HBA, and syringic acid was carried out using anion exchange chromatography with electrospray ionization and tandem mass spectrometric detection in negative ion mode (IC-ESI-MS/MS). This method

provides greater chromatographic selectivity for organic anions than HPLC, and better sensitivity due to the ability to inject larger samples on-column.

The analytical system consisted of a Dionex AS-AP autosampler, ICS-2100 integrated reagent-free ion chromatograph, and ThermoFinnigan TSQ Quantum triple quadrupole mass spectrometer. A 1 ml injection loop was used to introduce sample into the ion chromatograph. The separation was carried out on an IonPac AS18-Fast 2 $\mu$m analytical column with a 40 mM potas-

sium hydroxide eluent at a flow rate of 200 $\mu$l/min. Cations were removed from the column effluent using an electrolytically regenerated suppressor. Methanol was added to the eluent stream downstream of the suppressor in order to maintain a stable electrospray (70 $\mu$l/min, J.T. Baker LC-MS grade). The electrospray source was operated at -3 kV, with a sheath gas pressure of 38 psi, an auxiliary gas pressure of 15 psi, and an ion inlet cone temperature of 350°C. The analytes were detected in negative ion mode using a collision energy of 30 eV, using mass transitions: VA, m/z 167→108 and p-HBA, m/z 137→93. External

standards were prepared using pure VA and p-HBA (Sigma-Aldrich). Stock solutions of 100 ppb were prepared weekly and diluted daily to prepare working standards ranging from 0.1-2 ppb. VA and p-HBA were detected at retention times of 11.9 and 12.4 minutes with peak widths at half height of 0.4 minutes (Fig. 2). Limits of detection were estimated as 0.01 and 0.05 ppb, respectively, defined as 3 times the standard deviation of the MilliQ water blank. Note that the limit of detection was determined not by the sensitivity of the instrument but by our ability to generate and analyse laboratory blanks with lower levels of these compounds. The analytical system also was configured to detect syringic acid, with a retention time of 11.3, and a limit of detection of 0.09 ppb.

## 3   Results

### 3.1   Analytical results and data processing

In this study, VA concentrations are reported for 3,294 Akademii Nauk ice core samples, and 2,585 of these were also analysed for p-HBA (Fig. 3). The instrument was originally optimized to analyze VA. Several samples were analyzed for VA before a method was developed analyze p-HBA. In addition, a subset of 1,074 Akademii Nauk samples were analysed as replicates for vanillic acid using the older HPLC-ESI/MS/MS method described by Grieman et al. (2015). The results indicate no bias between the techniques and illustrate the improved sensitivity ($\sim$10 times better signal-to-noise) of the IC-based method (Fig. S3). For the remainder of this paper, only the IC-ESI/MS/MS results are discussed.

VA and p-HBA were below detection in 56% and 76% of the samples, respectively, and the frequency distributions of both compounds were skewed towards lower concentrations. Skewness is expected because burning is episodic and spatially heterogeneous, and atmospheric levels of burning aerosols are highly enriched during those episodes. The distribution was successfully normalized using a logarithmic transformation. Outliers were excluded from the entire dataset prior to transformation. Outliers were defined as VA and p-HBA levels outside $\pm 2\sigma$ from the median (0.0075 ppb and 0.021 ppb, respectively). This process excluded 0.43% of the VA data and 2.6% of the p-HBA data. For the entire Akademii Nauk ice core, the geometric means of VA and p-HBA were $0.0087^{+0.037}_{-0.0071}$ and $0.019^{+0.048}_{-0.014}$ ppb ($\pm 1\sigma$), respectively.

960 samples were analysed for syringic acid. Only 0.21% of these samples were above the detection limit, despite the fact that the detection limit of the instrument for syringic acid was similar to that of the other aromatic acids. Standard additions of syringic acid to ice core samples at concentrations comparable to the ambient levels of VA and p-HBA were recovered quantitatively, indicating that there was no suppression of signal due to matrix effects. The low levels of syringic acid suggest that it was either: 1) not generated at the biomass burning sources that impact the Akademii Nauk ice core, 2) chemically lost from the aerosols during transport, or 3) degraded in the ice core after deposition. Syringic acid is similar in molecular structure to the other two compounds, and it does not differ greatly in terms of volatility or reactivity. It is therefore most likely that syringic acid was not generated at the biomass burning sources. Syringic acid is structurally related to the lignin commonly found in grasses, including tundra grasses, and its absence in the ice core may simply indicate that grasses were not a significant component of the parent fuels (Oros et al., 2006). Laboratory studies of biomass burning indicate that syringic acid is not a

component of burning-derived aerosols from conifers (Iinuma et al., 2007; Otto et al., 2006; Rogge et al., 1998; Simoneit et al., 1999). We are not aware of any laboratory combustion studies of plant species typical of Siberian forests or tundra.

## 3.2 The Akademii Nauk vanillic and p-hydroxybenzoic acids time series

The raw time series of VA and p-HBA exhibit broadly similar patterns, each showing several multi-century periods of elevated levels (Fig. 3). Throughout the preindustrial Late Holocene (prior to 1700 CE), the levels of VA generally are higher than those of p-HBA. The median and mean of the ratio of VA to p-HBA prior to 1700 CE were 1.4 and 4.4, respectively. Elevated levels of both compounds also occur during the industrial period (after 1700 CE), but here the levels of p-HBA exceed those of VA. There are also numerous smaller multi-decadal features in both records, as well as higher frequency (sub-annual and inter-annual) variability throughout the ice core.

The Akademii Nauk site experiences summer-time surface melting and infiltration, which results in redistribution of soluble compounds of over 1 m of surface snow (roughly 2-3 annual layers; Fritzsche et al., 2005; Opel et al., 2009, 2013). The distribution of melt layers in the core over the last 500 years does not correlate with aromatic acid levels and is not likely to be responsible for the major features in the record (Fig. S4). Due to this disturbance in annual layering, we focus exclusively on multi-decadal and longer time scales. To remove short-term variability, smoothed records were constructed using log-transformed 40-year bin averaged VA and p-HBA records. The exponentials of the smoothed log-transformed records were used to present the records in concentration units (Fig. 4). Smoothing was also carried out using locally weighted polynomial regression (LOESS; Cleveland and Devlin, 1988). Bin-averaged and LOESS smoothing of the Akademii Nauk organic acid records show essentially the same centennial to millennial-scale features that are clearly visible in the raw data (Fig. S5). As evident from visual inspection of the bin-averaged record, VA and p-HBA are correlated ($r^2$=0.47, $p < 10^{-6}$, n=80; Fig. 5). The similarity in VA and p-HBA suggests that the two compounds are derived from a common source and/or are modulated by similar depositional/post-depositional processes.

For the purposes of this study, we define elevated periods as including data for which the standard error bands for 40-year binned averages reach or exceed the upper quartile of the entire dataset (Fig. 4). The start/stop dates for these time ranges each have an uncertainty of $\pm$ 20 years due to the bin averaging of the data. These time ranges are highly uncertain prior to 743 CE due to the high uncertainty of the age scale. Three periods of elevated VA are identified (650-300 BCE, 340-660 CE, and 1460-1660 CE), all of which are shared by p-HBA. p-HBA has an additional period of elevated levels from 1780-1860 CE that is not shared by VA. The levels of aromatic organic acids during this elevated period are enriched many-fold over the intervening "quiet" periods. This large dynamic range is quite different from the enrichment patterns typically found for inorganic ice core burning tracers such as ammonium and nitrate, where relatively small burning signals are superimposed on a large background from other natural sources such as biogenic emissions or lightning (Eichler et al., 2011; Fuhrer et al., 1993, 1996; Legrand et al., 1992, 2016; Legrand and De Angelis, 1996; Rubino et al., 2015; Savarino and Legrand, 1998; Taylor et al., 1996; Whitlow et al., 1994; Yalcin et al., 2006). We interpret these large pulses of aromatic acids in the Akademii Nauk ice core as evidence of multi-century periods of enhanced deposition of biomass-burning derived aerosols.

A period of elevated p-HBA levels is identified from 1780-1860 CE, using 40-year bin averaging of the entire dataset. This period is qualitatively different from the other elevated intervals in that p-HBA is more abundant than VA. The log-transformed dataset (after 1700 CE) was 10-year bin averaged to determine shorter-term elevated periods during the industrial period (1750-2000 CE). Elevated intervals during the industrial period are defined as periods during which the standard error bands of the

10-year binned averages reach or exceed the upper quantile of the dataset after 1700 CE (Fig. 6). VA and p-HBA are both elevated from 1860-1900 CE, using this definition. p-HBA is additionally elevated from 1770-1790 CE. The start and end dates of these periods each have an uncertainty of $\pm$ 5 years due to the application of bin-averaging.

VA and p-HBA may be subject to post-depositional revolatilization or photochemical destruction (Grannas et al., 2007). If post-depositional processes were a significant factor in determining the trends in aromatic acid levels in the ice core, one

might expect to see a strong relationship with accumulation rate or with ice chemistry. There is no evidence of large changes in accumulation that can explain the large concentration changes apparent in the VA and p-HBA measurements. The age-depth relationship for Akademii Nauk indicates constant average accumulation rate from 1700-1999 CE (140-0 m depth; Fig. S2). Below 140 m, the age-depth curve varies smoothly in a manner consistent with thinning due to ice flow at relatively constant accumulation (Opel et al., 2013). Similarly, there are no obvious correlations between major ion chemistry (seasalt-

derived $Na^+$, terrestrial-derived $Ca^{2+}$, or volcanic S) and the aromatic acid levels (Fig. S6). One could perhaps argue that acidification of aerosols and/or ice during the past century was responsible for the decline in p-HBA levels around 1900 CE due to revolatilization (Fig. 6). However, large volcanic sulfate peaks throughout the record do not exhibit evidence of loss of aromatic acids. Given the absence of indication of any postdeposition artifacts, we interpret ice core VA and p-HBA as tracers for biomass burning variability. Further investigation of such effects is warranted, particularly for low accumulation ice core

sites. Future studies should examine the relationship between levels of aromatic acids in air and snow, in order to develop transfer functions following the method of Fischer et al. (2015).

## 4   Discussion

### 4.1   Potential source regions and vegetation types – air mass back trajectories

The possible source locations for biomass burning impacting the Akademii Nauk ice core site were identified by calculating

the fraction of air mass back trajectories from the ice core site originating in or passing over Siberia (defined as east of 42°E), Europe (defined as west of 42°E), or North America. This analysis assumes present-day meteorological conditions. Changes in atmospheric circulation patterns over the past millennia may have occurred, but are not considered here. Each region was subdivided into ecofloristic zones defined by the Food and Agriculture Organization (Fig. S7; http://cdiac.ornl.gov/epubs/ndp/global_carbon/carbon_documentation.html; Ruesch and Gibbs, 2008). Air mass back trajecto-

ries were computed using the HYSPLIT model (Draxler and Hess, 1997, 1998; Draxler et al., 1999; Stein et al., 2015). 10-day back-trajectories from the ice core site (80°N, 94°E) were started at 100 m above ground level, at 12:00 AM and 12:00 PM local time (UTC + 7 hours) each day for spring (trajectories beginning March 1-May 31), summer (trajectories beginning June

1-August 31), and fall (trajectories beginning September 1-November 30). NCEP/NCAR Reanalysis data from 2006-2015 CE was used (ftp://arlftp.arlhq.noaa.gov/pub/archives/reanalysis; Kalnay et al., 1996).

The results indicate that Siberia is the most likely source region to the Akademii Nauk ice core site, with most of the trajectories either originating in or transecting this region (Table 1; spring 61%, summer 28%, and fall 60%). Siberian boreal tundra woodlands, boreal coniferous forests, and boreal mountain systems all contributed significantly (Fig. S8). There were some seasonal differences, with more trajectories originating from or passing over these areas in the spring and fall than in the summer. All of the other ecofloristic zones in Siberia intersected fewer than 5% of the trajectories. Similarly, for these 10-day back trajectories, all of the ecofloristic zones in Europe and North America contributed less than 3% for all three seasons. This analysis does not prove that burning emissions from Europe and North America could not contribute to the ice core signals, but that they would require significantly longer atmospheric transport times.

Both the trajectory analysis and the VA/p-HBA ratio in the ice core are consistent with Siberian conifer forests and tundra woodlands as main sources for the three major preindustrial burning peaks in the Akademii Nauk record. Laboratory studies show that burning conifers results in higher or similar yields of VA to p-HBA (Iinuma et al., 2007; Oros and Simoneit, 2001a; Otto et al., 2006; Rogge et al., 1998; Simoneit et al., 1999). Pine wood burned in a fire in Northern Alberta, Canada yielded a VA/p-HBA ratio of 27:4 $\mu$g/g carbon (Otto et al., 2006). European pine combustion yielded a VA/p-HBA ratio of 14:1.6 mg/kg of fuel burned (Iinuma et al., 2007). Burning of Ponderosa pine, Sitka spruce, and Douglas fir yielded VA/p-HBA ratios of 790:40, 2194:2968, and 3441:4345 $\mu$g/kg of fuel burned, respectively (Oros and Simoneit, 2001a).

In the burning peak after 1700 CE, the levels of p-HBA are higher than VA. This likely indicates a change in the type of vegetation burned. This could reflect either a change in ecosystem at the source region, a shift in the location of burning, or a change in atmospheric transport pattern. Laboratory studies indicate that tundra grass fires yield high p-HBA to VA ratios. We therefore speculate that Siberian tundra fires or boreal peat fires may have contributed to high levels of aromatic acids during this period (Iinuma et al., 2007; Oros et al., 2006).

The Akademii Nauk aromatic acid record suggests that high biomass burning emissions were sustained for multi-century periods during the last 2,600 years of the Holocene. This result perhaps indicates that the fires were widespread, but of relatively low intensity, consistent with the fact that low intensity ground fires are the principle mode of burning in Eurasian boreal forests today. Model results suggest that ground fires dominated the Eastern Siberian region over the past 1,200 years (Ito, 2005). By contrast, North American boreal forest burning occurs predominantly by high intensity, stand-replacing crown fires (Ito, 2005; Rogers et al., 2015; Stocks et al., 2001; Ryan, 2002). Aromatic acids should be measured in ice cores from North America to examine if the fire conditions are reflected in the pattern of centennial-scale variability.

## 4.2   Comparison to other biomass burning proxy records

There are few records of Siberian biomass burning covering the past 2,600 years, and none of the existing records show all three of the prominent periods of increased fire activity present in the Akademii Nauk record (Fig. 7). Measurements of nitrate, potassium, and charcoal in a 750-year ice core from southern Siberia suggest elevated burning from 1600-1680 CE (Eichler et al., 2011). This period overlaps the most recent of the major preindustrial peaks in the Akademii Nauk VA record

(1460-1660 CE). A number of organic biomass burning tracers were measured in a 300-year ice core from the Kamchatka Peninsula, Northeast Asia (Kawamura et al., 2012). In that study, elevated p-HBA, VA, dehydroabietic acid, and levoglucosan were observed during the periods from 1700-1800 and 1880-2000 CE. Akademii Nauk p-HBA and VA are similarly elevated from 1770-1790 (p-HBA only) and 1860-1900 CE. VA and p-HBA remain elevated late in the $20^{th}$ century in the ice core from the Kamchatka Peninsula. It is not possible to determine if Akademii Nauk VA and p-HBA also increase during this period due to limited Akademii Nauk sample availability after 1970.

Historical changes in biomass burning have also been reconstructed from ammonium spikes in the NEEM and Summit Greenland ice core records. Variations in the frequency of ammonium peaks for the past 1,000 years suggest elevated burning from 1200 -1500 CE and low burning from 1600-1800 CE (Legrand et al., 2016; Fischer et al., 2015). This pattern of preindustrial burning is different from the Akademii Nauk record, which is not surprising given that Greenland is primarily influenced by transport from North America, rather than Eurasia. These trends are generally consistent with charcoal records from northeastern Canada (Power et al., 2013). Zennaro et al. (2014) presented a 2,000-year NEEM record of levoglucosan and black carbon. They show four preindustrial maxima in levoglucosan around 100 BCE-100 CE, 200-600 CE, 1000-1200 CE, and 1500-1700 CE. The last of these maxima is strongest and coincident with the 1400-1600 peak in Akademii Nauk aromatic acids. Interestingly, the same feature is not the largest peak for NEEM black carbon. There are clearly unresolved differences between various Greenland ice core proxy records, particularly for the period around 1500 CE.

As noted earlier, there are only 11 charcoal records from the Siberian region in the Global Charcoal Database (Blarquez et al., 2014). Of these records, only the record from Bolshoe Bog in the Lake Baikal region of southern Siberia has sufficient temporal resolution that allows comparison with the Akademii Nauk record. The Bolshoe Bog record does exhibit similar timing of elevated levels to Akademii Nauk (Fig. 8), suggesting a common biomass burning source region between the two records. Further charcoal studies throughout Siberia are needed to assess the full range of biomass burning source regions contributing to aromatic acids in the Akademii Nauk ice core.

## 4.3 Comparison to climate proxy records

Variability in biomass burning can be caused by human activity. The effect of humans on Siberian wildfire activity is not well-established. Human civilizations were predominantly nomadic in Siberia prior to $16^{th}$ century. Comparison between pollen records, civilization development, and climate in the Lake Baikal region suggests that vegetation changes were more likely linked to climate than human-induced land use change throughout the Holocene (Tarasov et al., 2007).

Variability in regional biomass burning generally is driven by changes in temperature and precipitation, which are linked to atmospheric circulation patterns. Over recent decades, Siberian wildfire burned area correlates with changes in the Arctic Oscillation, with increased biomass burning during the positive phase of the Arctic Oscillation when Siberian summers are warmest (Balzter et al., 2005, 2007; Seki et al., 2015). An 8,000-year Holocene proxy record of Arctic Oscillation shows a 1,500-year cycle (Darby et al., 2012), but it is not synchronous with increased biomass burning in the Akademii Nauk ice core. This sedimentary proxy evidence does not support the Arctic Oscillation as the primary mode of climate variability controlling Siberian burning on millennial time scales.

Eichler et al. (2011) concluded that the period of elevated burning recorded in the Belukha glacier was preceded by a drought event that was possibly related to the positive phase of the Pacific Decadal Oscillation (PDO). The period of elevated VA and p-HBA from 1460-1660 CE also overlaps a positive phase of the PDO reconstructed using tree ring chronologies (MacDonald and Case, 2005). A longer record of the PDO is needed to determine if the other peaks in the Akademii Nauk VA and p-HBA records follow this pattern. Zennaro et al. (2014) also relate the period of elevated burning in the NEEM Levoglucosan record from 1500 to 1700 CE to drought conditions. They link Asian drought conditions to monsoon failures during the $16^{th}$ and $17^{th}$ centuries. This variability may also be related to the PDO, given that the PDO can modulate the summer monsoon (Chen et al., 2013; Krishnamurthy and Krishnamurthy, 2014).

Climate reconstructions based on Northern Hemisphere proxy records show a long-term cooling trend over the past 2,000 years (Fig. 8; PAGES 2k Consortium, 2013; Hegerl et al., 2006; Ljungqvist, 2010; Mann et al., 2008; Marcott et al., 2013; Moberg et al., 2005). Centennial-scale climate variability, most notably the Medieval Climate Anomaly (830-1100 CE) and Little Ice Age (1580-1880), is superimposed on this trend (PAGES 2k Consortium, 2013; Büntgen et al., 2016; Lamb, 1965; Mann et al., 2009). Akademii Nauk VA and p-HBA levels do not exhibit a trend following the long-term cooling trend in temperature, but they do appear to correlate with some centennial-scale climate variability. Akademii Nauk VA and p-HBA levels are elevated from 340-660 CE, prior to and during the Late Antique Little Ice Age (536-660 CE; Büntgen et al., 2016). The Akademii Nauk aromatic acids are low during the early part of the Medieval Climate Anomaly (prior to 1050 CE). They are slightly elevated during the latter part of the Medieval Climate Anomaly. Tree ring reconstructions suggest that the Medieval Climate Anomaly was humid in Northern Siberia (Sidorova et al., 2013). Akademii Nauk VA and p-HBA are elevated from 1460-1660 CE during the Little Ice Age. The Akademii Nauk aromatic acid trends are different from those in composite Northern Hemisphere sedimentary charcoal records, which show an overall decline over the past 2,000 years, with a maximum during the Medieval Climate Anomaly and minimum during the Little Ice Age (Marlon et al., 2008; Power et al., 2013).

The Little Ice Age is the most recent in a series of Holocene cooling events known as "Bond events" (Bond et al., 1997, 1999). Bond events are episodes of increased ice rafted debris in North Atlantic sediment cores throughout the Holocene at intervals of 1,470±500 years with durations from 200-500 years. Bond events may be the result of a combination of ~1,000-year and ~2,000-year cycles of climate variability in the Holocene (Obrochta et al., 2012). The three most recent Bond events were centered at: 2,800, 1,400, and 500 years before present (850 BCE, 550 CE, and 1450 CE; Fig. 8; Bond et al., 1997). The two most recent major periods of increased Siberian burning found in the Akademii Nauk ice core are similar in timing to the two most recent Bond Events. The earliest major peak in the Akademii Nauk ice core is later than the Bond Event centred at 850 CE. This difference in timing could be due to the uncertainty in the Akademii Nauk ice core age scale. Simultaneous changes in climate also are observed in the Chinese speleothem record from Dongge Cave, centered at: 2,700, 1,600, and 500 years before present (750 BCE, 350 CE, and 1450 CE; Fig. 8, bottom plot; Wang et al., 2005). The speleothem record shows three 100-500 year periods of increased $\delta^{18}$O, indicating decreased East Asian summer monsoon intensity (Dykoski et al., 2005; Hu et al., 2008; Wang et al., 2005; Yuan et al., 2004).

The East Asian winter monsoon is affected by changes in the intensity of the Siberian High (Gong et al., 2001; Wu and Wang, 2002). Comparisons between temperature and precipitation records between 1922 and 1999 CE show that when the

Siberian High became stronger, temperatures and precipitation over the Eurasian continent decreased (Gong and Ho, 2002). Drier conditions resulting from variability of the Icelandic Low and the Siberian High could have altered biomass burning. The long-term variability of the Siberian High is not well-established (D'Arrigo et al., 2005), but GISP2 ice core sea salt Na (ssNa) and non-sea salt K (nssK) records indicate that the Icelandic low (ssNa) and the Siberian high (nssK) were stronger than mean levels when VA and p-HBA were elevated between 1460-1660 CE (Meeker and Mayewski, 2002). The Akademii Nauk data support the suggestion by Zennaro et al. (2014) of a link between northern hemisphere boreal fires and monsoon weakening. The similarity in timing between the Siberian biomass burning pulses, the Bond events, and the monsoonal changes likely suggests a link in this region between fires and large-scale climate variability on millennial time scales.

## 5 Conclusions

The Akademii Nauk VA and p-HBA measurements constitute the first millennial scale ice core record of aromatic acids. In the modern atmosphere, these compounds are predominantly associated with aerosols derived from boreal or high latitude biomass burning. Hypothetically, atmospheric aromatic acids may also originate in the form of direct biogenic or soil emissions, or from breakdown of mechanically transported soil humic compounds. There is yet no evidence showing that either of these processes is a significant atmospheric source. We therefore propose that the Akademii Nauk aromatic acid record is a historical record of biomass burning in Northern Eurasia. Because these compounds are deposited on the ice cap after long-distance transport in the atmosphere, the record likely is modulated by emissions and transport. Changes in vegetation types and in combustion conditions (temperature, humidity, oxygen levels, etc.) are other processes that may influence aromatic acid levels in ice cores. Until the impact of the multitude of contributing processes can be quantified, ice core aromatic acid records should be interpreted as qualitative biomass burning proxies.

Regional millennial-scale Siberian wildfire activity is not well-established due to a paucity of proxy records in the region. Siberian biomass burning may be linked to North Atlantic climate variability and the Asian monsoon. Regional records of Siberian precipitation changes would help to uncover how Bond Events may have affected climate in Siberia. More and longer records of the Arctic Oscillation and PDO are needed to reveal a relationship between Siberian biomass burning and atmospheric circulation. This study demonstrates that ice core records of organic compounds that are uniquely derived from biomass burning, such as aromatic acids, have the potential to add to our understanding of regional-scale trends in biomass burning and their relationship to climate.

## 6 Data availability

The data presented in this paper will be submitted to the NSF Arctic Data Center (https://arcticdata.io/) prior to publication.

*Author contributions.* Mackenzie Grieman analyzed the ice core samples for this study. Mackenzie Grieman and Eric Saltzman developed the analytical method, processed the data, and wrote the manuscript text. Murat Aydin helped to develop the analytical method and provided comments on the manuscript. Diedrich Fritzsche and Thomas Opel drilled and/or processed the Akademii Nauk ice core, developed the original depth-age scale, sent samples from the upper portion of the ice core to UC Irvine, and provided comments on the manuscript. Joe McConnell and Michael Sigl provided the ice core melter samples and ice core aerosol measurements, developed the updated depth-age scale, and provided comments on the manuscript.

*Competing interests.* The authors declare that they have no conflict of interest

*Acknowledgements.* We would like to acknowledge O. Masselli, D. Pasteris, and N. Chellman of the Desert Research Institute for assistance in analyzing the Akademii Nauk ice core and collecting the ice core melt samples, T. Sutterley for help with coding, and J.-Y. Yu and K. Johnson for helpful comments on the manuscript. This research was supported by a generous donation from the Jenkins Family to the Department of Earth System Science, University of California, Irvine and by the Antarctic Glaciology program of the National Science Foundation (ANT-0839122; PLR-1142517). The National Science Foundation also provided support to E.S.S. through the Independent Research and Development program. Analysis of the Akademii Nauk at the Desert Research Institute was funded by ARC-1023672. This work furthermore contributes to the Eurasian Arctic Ice 4k project (grant OP 217/2-1 by Deutsche Forschungsgemeinschaft awarded to T.O.).

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

**Table 1.** Fractions of air mass back trajectories originating from or intersecting various ecofloristic zones and geographic regions (% rounded to nearest integer). Ecofloristic zones are defined by Food and Agriculture Organization (Fig. S7; http://cdiac.ornl.gov/epubs/ndp/global_carbon/carbon_documentation.html; Ruesch and Gibbs, 2008).

| | | Season | | |
|---|---|---|---|---|
| Geographic region | Ecofloristic zone | Spring (%) | Summer (%) | Fall(%) |
| Siberia | Boreal tundra woodland | 36 | 7 | 40 |
| | Boreal coniferous forest | 23 | 3 | 32 |
| | Boreal mountain system | 17 | 2 | 22 |
| | Total | 61 | 28 | 60 |
| Europe | Total | <1 | 2 | <1 |
| North America | Total | 11 | 18 | 8 |

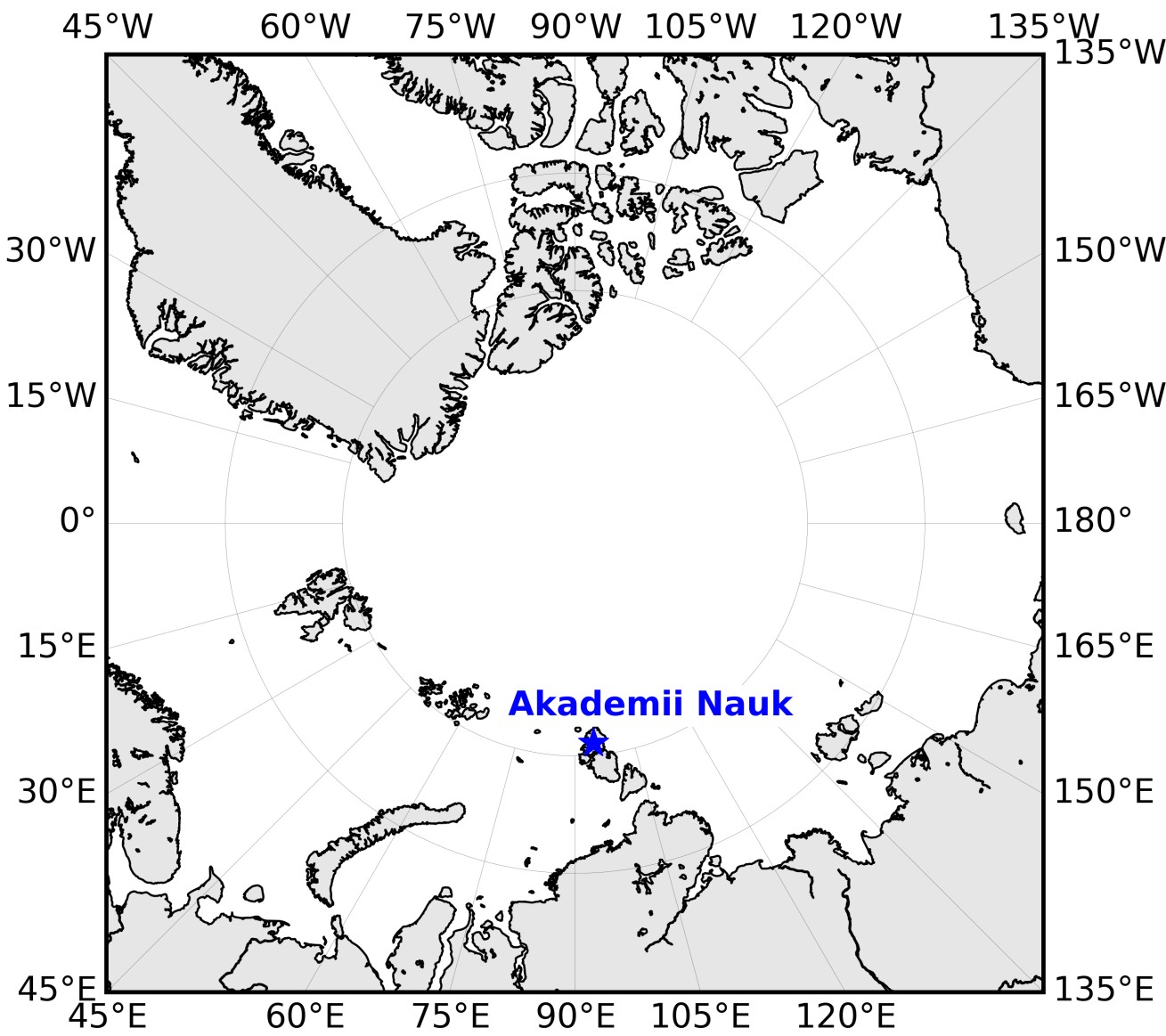

**Figure 1.** Location of Akademii Nauk ice core drilling site (80°31'N, 94°49'E).

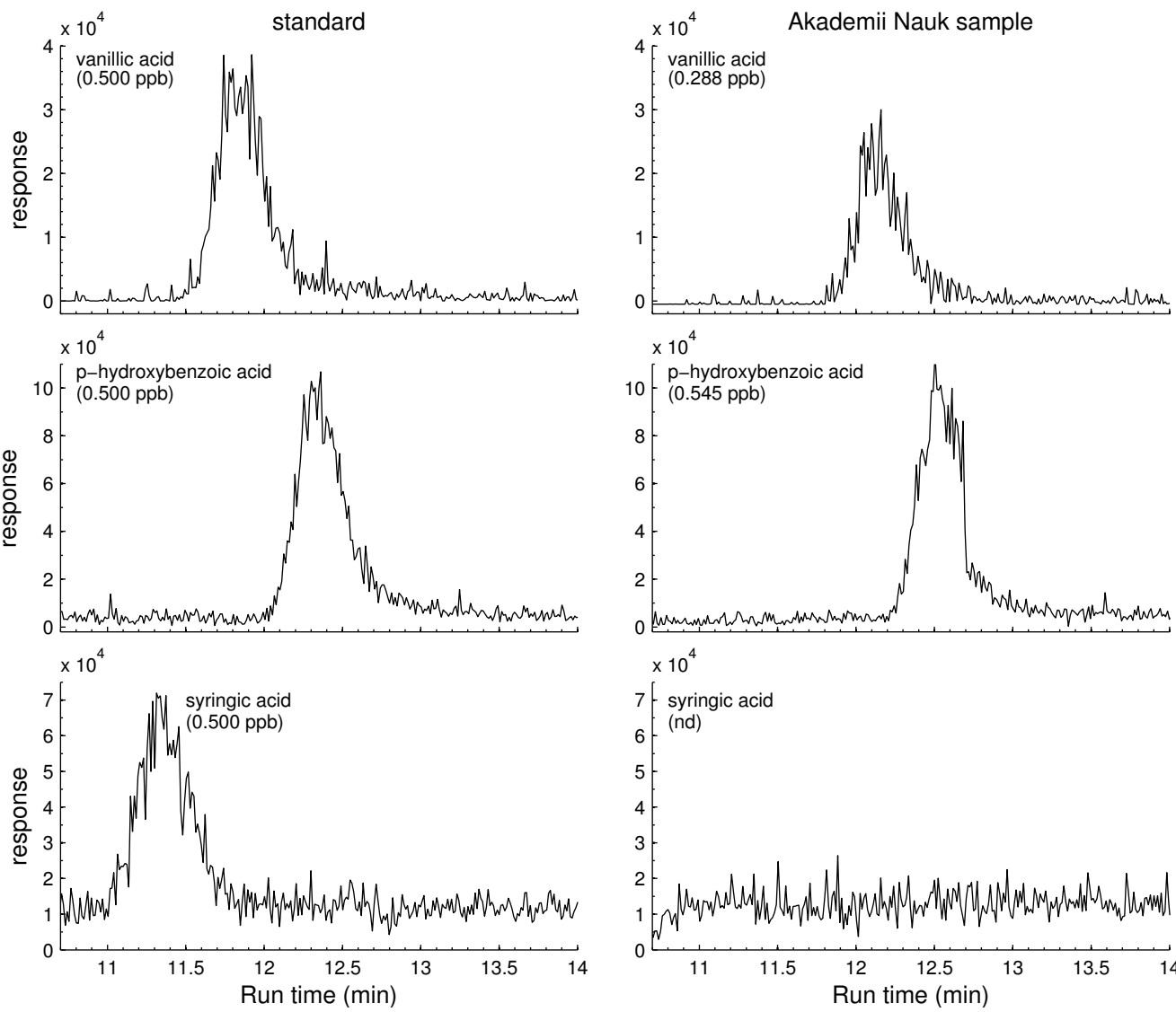

**Figure 2.** Analysis of vanillic acid, para-hydroxybenzoic acid, and syringic acid using ion chromatography with electrospray ionization and tandem mass spectrometry. Left: standards, Right: Akademii Nauk ice core sample (219 m, 1450 CE) containing 0.288 ppb vanillic acid and 0.545 ppb para-hydroxybenzoic acid. Syringic acid was not detected in the sample.

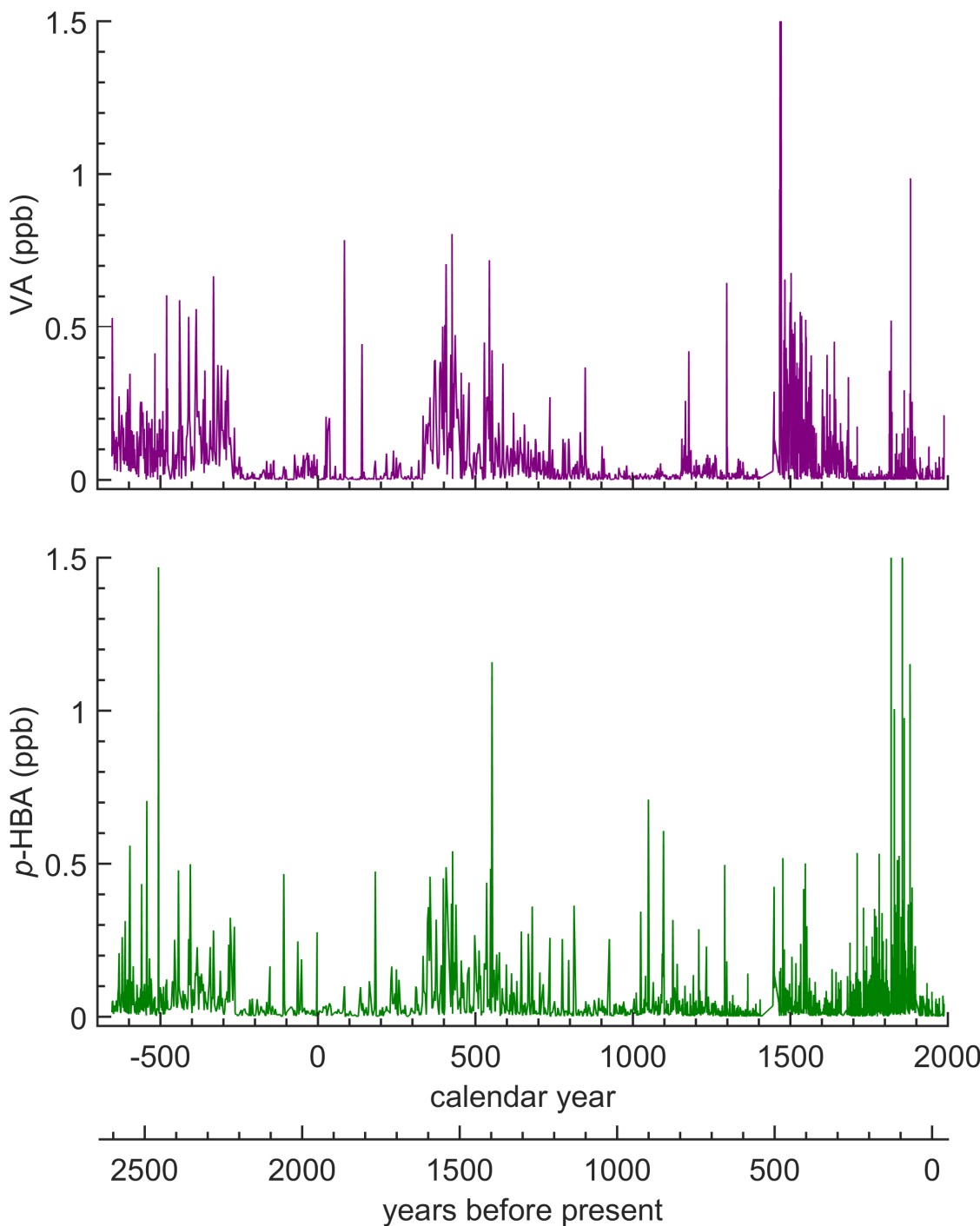

**Figure 3.** Akademii Nauk vanillic acid (top) and para-hydroxybenzoic acid (bottom) ice core records.

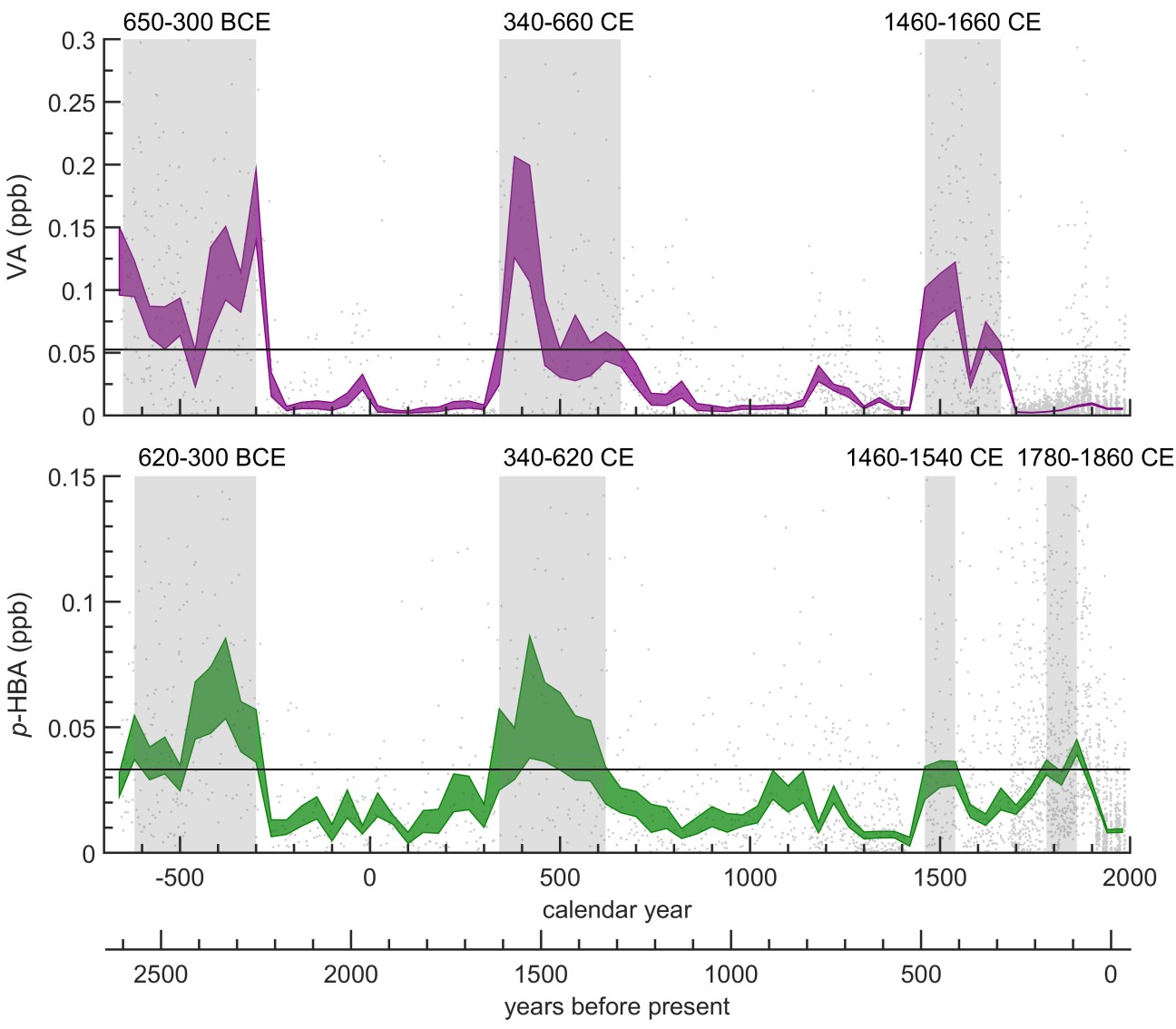

**Figure 4.** Akademii Nauk vanillic acid (top) and para-hydroxybenzoic acid (bottom) ice core records. Individual measurements are shown as grey dots. The colour-filled lines are exponentials of $\pm 1$ standard errors of 40-year bin averages of the log-transformed data. The solid horizontal lines represent the $75^{th}$ percentile of each dataset. The vertical gray shaded areas are periods of elevated vanillic acid or para-hydroxybenzoic acid, identified as periods when the bin-averaged data are in the upper quartile of the transformed dataset.

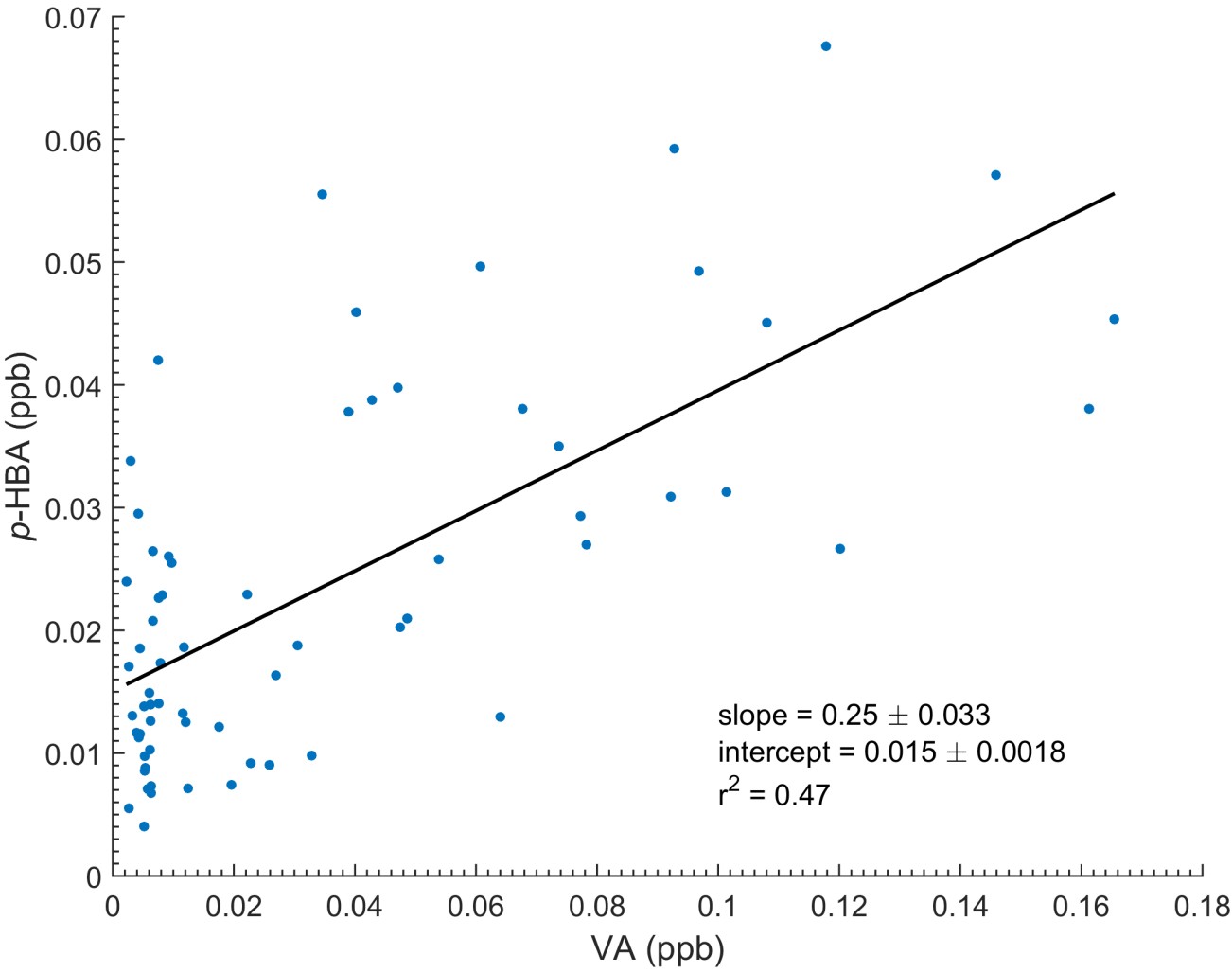

**Figure 5.** Comparison of vanillic acid and para-hydroxybenzoic acid records. Linear fit is the 40-year bin averaged log-transform of vanillic acid against the 40-year averaged log-transform of para-hydroxybenzoic acid.

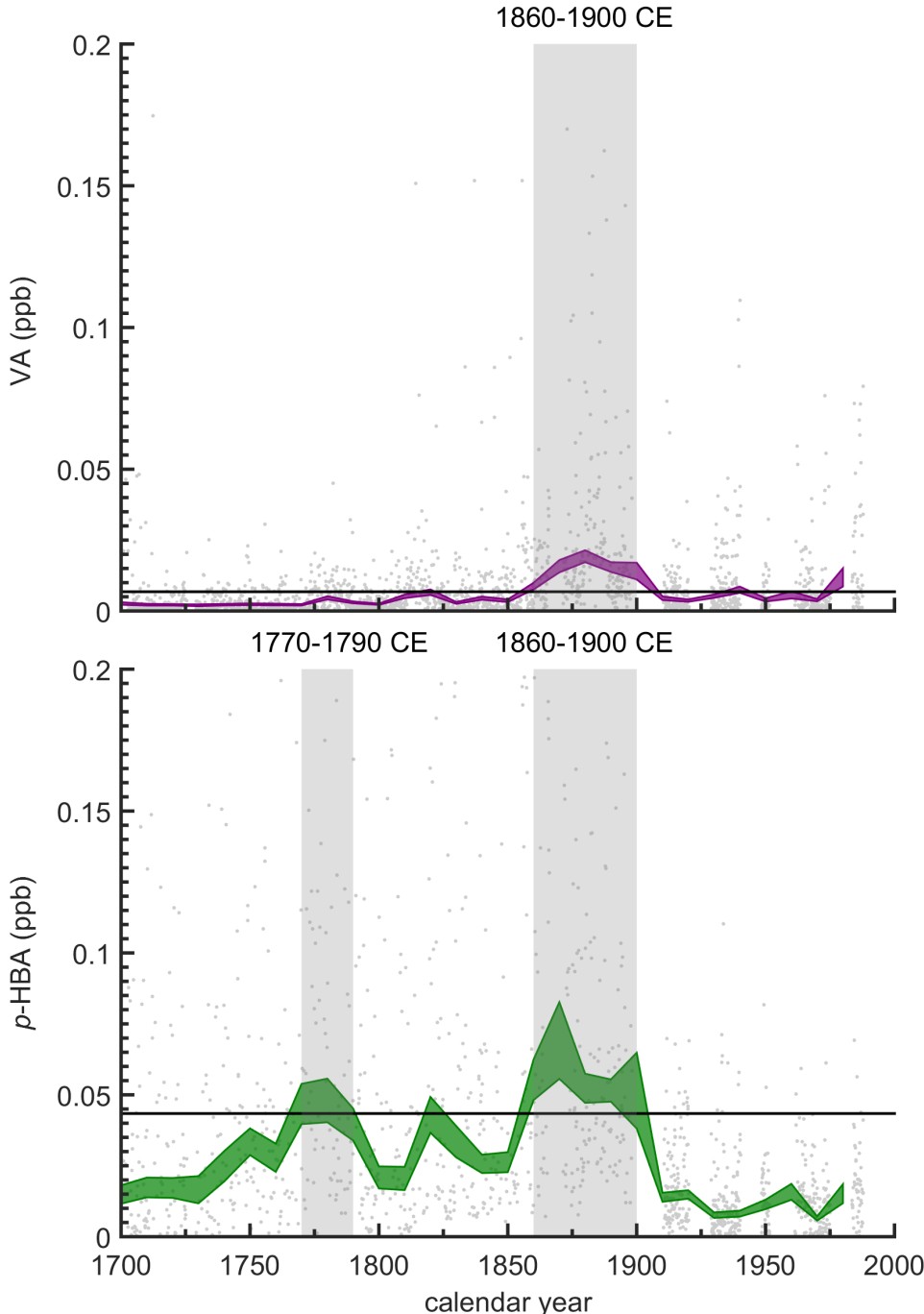

**Figure 6.** Akademii Nauk vanillic acid (top) and para-hydroxybenzoic acid (bottom) ice core records. Individual measurements are shown as grey points. The colour-filled lines are exponentials of $\pm 1$ standard errors of 10-year bin averages of the log-transformed data. The solid horizontal lines represent the $75^{th}$ percentile of each dataset (after 1700 CE). The vertical grey shaded areas are periods of elevated vanillic acid or para-hydroxybenzoic acid, identified as periods when the bin-averaged data are in the upper quartile of the transformed dataset.

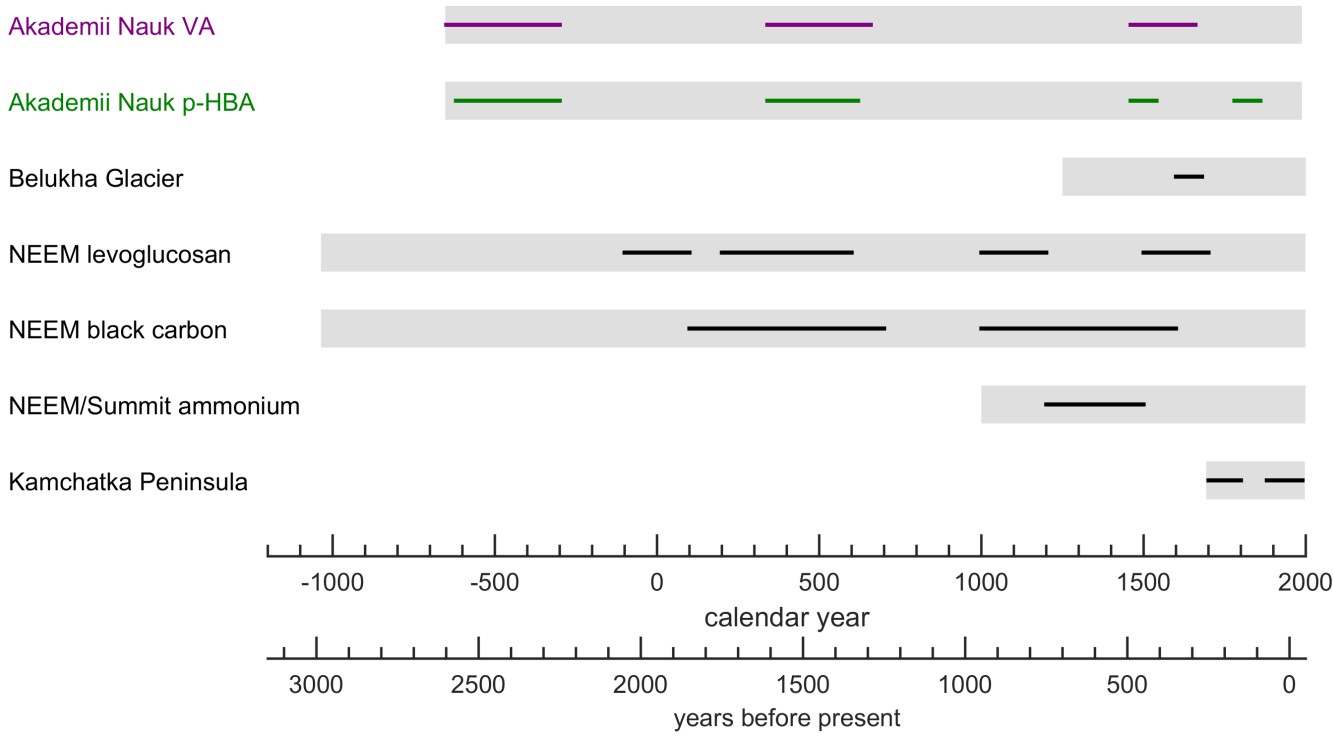

**Figure 7.** Timeline of elevated burning periods in Northern Hemisphere ice core studies. From top: Akademii Nauk vanillic acid and para-hydroxybenzoic acid (this study); Belukha glacier nitrate, potassium, and charcoal (Eichler et al., 2011); NEEM levoglucosan and black carbon (Zennaro et al., 2014); NEEM and Summit ammonium (Legrand et al., 2016; Zennaro et al., 2014); and Kamchatka Peninsula para-hydroxybenzoic acid, vanillic acid, dehydroabietic acid, and levoglucosan (Kawamura et al., 2012). Lines indicate periods of elevated burning. Grey bars mark the time range analysed in each core.

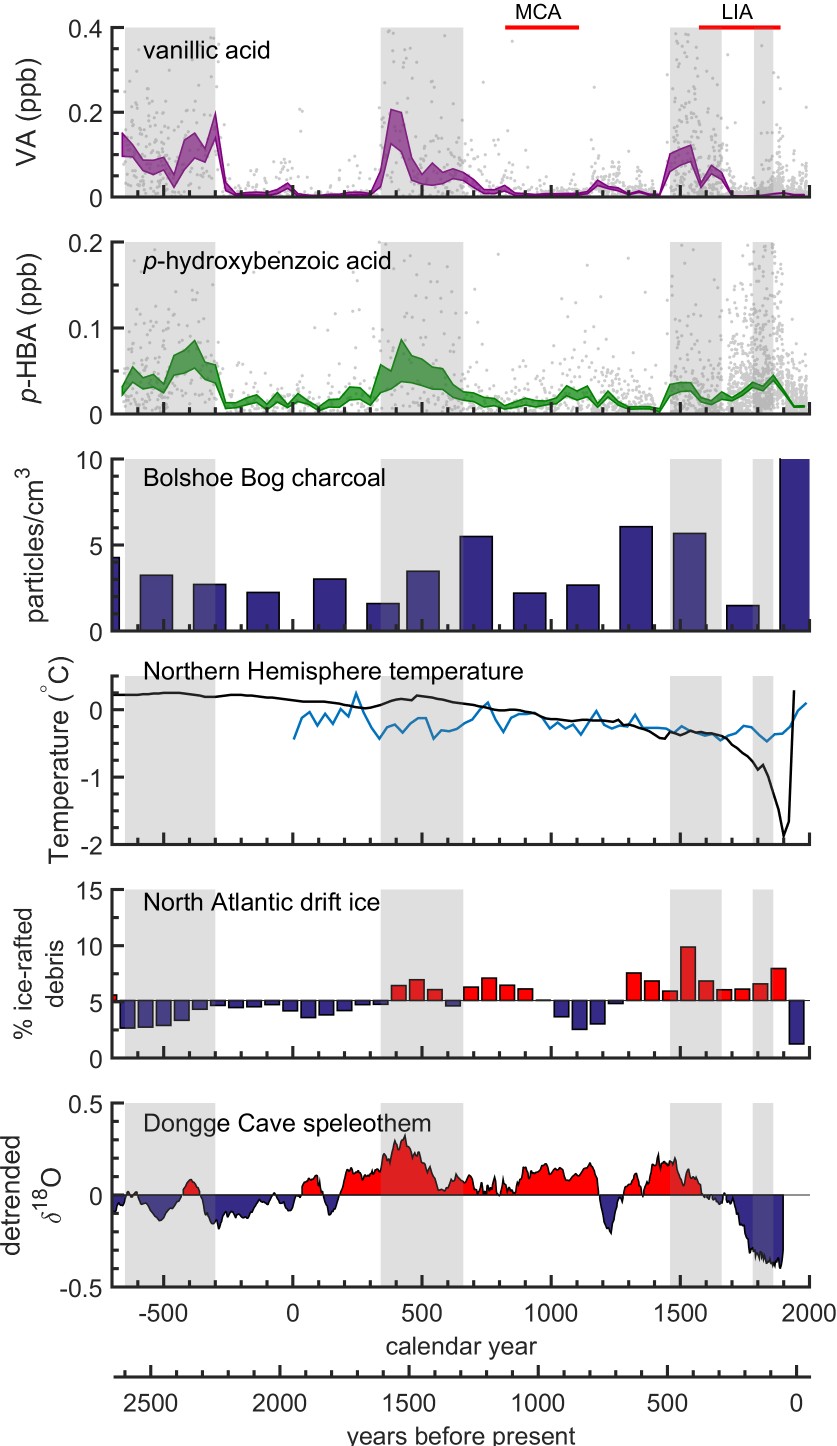

**Figure 8.** (Caption on next page.)

**Figure 8.** Comparison of the timing of aromatic acid signals in the Akademii Nauk ice core over the past 3,000 years compared to other climate-related proxy records. From top: 40-year bin-averaged (violet fill is 1 standard error of log transform) Akademii Nauk ice core vanillic acid measurements from this study, 40-year bin-averaged (green fill is 1 standard error of log transform) Akademii Nauk ice core para-hydroxybenzoic acid measurements from this study, Bolshoe bog charcoal record (Blarquez et al., 2014), 30-year medians of domain areas of PAGES 2k temperature reconstructions (blue; PAGES 2k Consortium, 2013) and 20-year means of zonal 30-90°N stacked temperature reconstruction (black; Marcott et al., 2013), North Atlantic ice-rafted debris indicating Bond Events (blue < mean; red > mean) (Bond et al., 2001), and smoothed Dongge Cave climate record from Asia showing changes in the monsoon using a moving average (blue < 0, red > 0; window size = 15) (Wang et al., 2005). The vertical grey shaded areas are periods of elevated vanillic acid and para-hydroxybenzoic acid, identified as periods when the bin-averaged data are in the upper quartile of the transformed dataset. The red lines are the Medieval Climate Anomaly (MCA) and the Little Ice Age (LIA).