# Peer review of "Aromatic acids in a Eurasian Arctic ice core: a 2,600-year proxy record of biomass burning"

_Climate of the Past, 2016_

## Referee Comment (RC4)

This paper reports on the concentration of aromatic acids (vanillic and parahydroxybenzoic) measured along a Eurasian Arctic ice core covering the last 3000 years. These acids that are used as proxies of biomass burning were measured by using a newly developed technique (IC-ESI-MS/MS). The obtained records are then compared to those from other ice cores extracted at sites potentially impacted by boreal forest fires including the Altai, the Kamchatka, and the Greenland ice cap.

Data on past frequency of boreal fires are of great importance since the boreal forest represents an important carbon reservoir and experiences predominantly natural fires of which the severity is expected to change with future warming and the subsequent modification of spring/summer conditions. In contrast to Canadian (or Alaskan) fires, Siberian fires are far less documented except for the very last decades when satellite data has strongly increased the accuracy of estimated burned area of this region. This paper that provides a rather unique record of Siberian fires over several past millennia is therefore of great interest for scientific communities working on forest fire records in ice cores and lake sediments as well as for the general topic of climate/fire conditions/vegetation interactions.

Overall the manuscript is already in a very good state (presentation of data, figures, scientific discussions, references) and my major comment is related to one statement made in the introduction (lines 20-24) and a paragraph within section 4.2 where authors compare their findings with Greenland ice core ones (see below). I therefore recommend publication of the manuscript (after authors consider the question rise below).

**Page 2, Line 20-24:** Please refer that this point has been extensively addressed in Legrand et al. (2006) and it was concluded that "for Greenland ice, ammonium, formate, OC (DOC or TOC), BC, as well as vanillic and glycolic acids were enhanced well above their background values during fire events. »

**Comments on Section 4.2:**

Line 24-29, Page 9 (also lines 15-18, page 10): This paragraph needs to be reworded since, as it stands, it gives to the reader the overall impression that only levoglucosan records in Greenland ice are available and useful to discuss past biomass burning activity, which is not correct. In fact, the NEEM record (5-year average) of levoglucosan (Zennaro et al., 2014) suggests an outstanding maximum around 1600 (not revealed by the sub-annual BC and ammonium profiles). At the opposite, the two NEEM and the Summit high-resolution ammonium records consistently indicate high fire activity from 1200 to 1500 and after 1850, whereas low fire activity occurred particularly from 1600 to 1800. These changes are also consistent with a composite series of charcoal records related to northeast boreal fires in North America were obtained (Power et al., 2012), confirming the high fire activity at the transition from the 19th to 20th century and to a lesser extent during the Medieval Warm Period, and the very low fire activity at the end of the Little Ice Age.

So making a too strong point on the past fire activity derived from a levoglucosan Greenland ice record would give the feeling that Greenland (in contrast to what tell us the air mass backward trajectories) better records Siberian than

Canadian fires. Concerning the levoglucosan, first as mentioned line 2 (page3)("the utility of this compound as a quantitative tracer is somewhat controversial due to the potential for rapid degradation in the atmosphere (Hoffmann et al., 2010; Hennigan et al., 2010; Slade and Knopf, 2013) ». Second, I just would like to emphasize that the comparison between the Greenland ice levoglucosan record and the Akdemii Nauk one is a bit misleading since your record shows numerous events with elevated level of aromatic acids (your figure 3) within the 1460-1660 CE time period for instance, whereas the « corresponding » levoglucosan Greenland peak is made of one or two outstanding values.

Therefore, I would like to propose a more adequate wording in updating the paragraph as follows (it is of course a suggestion):

"Ammonium ice records that consistently indicate Canada as the main source for fire plumes reaching Greenland also suggest changes in fire activity in response to climatic fluctuations over the last millennium (Legrand et al., 2016 and references therein). Indeed, the ammonium records from two NEEM and one Summit ice core reveal high fire activity from 1200 to 1500 and after 1850, whereas low fire activity occurred particularly from 1600 to 1800. These Greenland records suggest temporal changes coinciding fairly well with the occurrence of the warm and dry climate of the Warm Medieval Period (MWP; 1200–1350) and the cold climate of the Little Ice Age (LIA; 1600–1830). These past changes are consistent with composite series of charcoal records related to northeast boreal fires in North America (Power et al., 2012) indicating a high fire activity at the transition from the 19th to 20th century and to a lesser extent during MWP, and a very low fire activity at the end of the LIA. If correct, that suggests a different response of the Siberian and Canadian fire activity over the last millennium. To date, the NEEM record (5-year average) of levoglucosan (Zennaro et al., 2014) suggests a different picture for past changes with an outstanding maximum around 1600 (not revealed by the sub-annual BC and ammonium profiles). But, as discussed by Legrand et al. (2016), further works are here needed to understand the cause of the observed difference in past fire activity changes derived from levoglucosan, black carbon, and ammonium Greenland ice records."

Power, M. J., Mayle, F. E., Bartlein, P. J., Marlon, J. R., Anderson, R. S., Behling, H., Brown, K. J., Carcaillet, C., Colombaroli, D., Gavin, D. G., Hallett, D. J., Horn, S.P., Kennedy, L. M., Lane, C.S., Long, C. J., Moreno, P. I., Paitre, C., Robinson, G., Taylor, Z., and Walsh, M. K.: Climatic control of the biomass-burning decline in the Americas after AD 1500, The Holocene, 23, 3–13, doi:10.1177/0959683612450196, 2013.

End of the review

---

## Short Comment (SC1) · 24 Dec 2016

This is an important contribution which adds a valuable new palaeoclimate data point in a remote area. The low wildfire phase during the Medieval Climate Anomaly (MCA) matches well with other studies from the region which reported a switch towards more humid conditions during the MCA in northern Siberia and northern Europe:

Andreev et al. 2003: Levinson-Lessing Lake, Taymyr Peninsula Warm and wet phase 700-1200 AD http://onlinelibrary.wiley.com/doi/10.1111/j.1502-3885.2003.tb01230.x/abstract

Sidorova et al. 2013: Eastern Taimyr peninsula "The Medieval Warm

[Figure]

Period was wetter compared to 4111–3850 BC and 1791–2008 AD."
http://www.sciencedirect.com/science/article/pii/S0277379113001856

Wolfe et al. 2000: Middendorf Lake, western Taimyr Peninsula Medieval Climate
Anomaly 800-1200 AD characterized by more negative d18O interpreted as more hu-
mid climate http://www.sciencedirect.com/science/article/pii/S0033589400921240

For wet MCA hydroclimate in northern Europe see green data points on this clickable
MCA online map: http://t1p.de/mwp

---

## Referee Comment (RC1) · Anonymous Referee #1 · 29 Dec 2016

The manuscript of Grieman et al describes the quantitative analysis of aromatic acids (particularly vanillic acid and para-hydroxybenzoic acid) from a Eurasian Arctic ice core, covering a span of >3000 years before present. These acids provide a tracer for biomass burning, due to their sourcing from lignin combustion. The analytical method used is unique, and the data generated are compared to an existing technique with good agreement (with the new method providing a lower limit of detection for the compounds being studied). Evidence of biomass burning events are presented and the record from this core is compared to other proxies (of the few available) with relatively good qualitative agreement. The data provide an important addition to the literature, particularly given the scant information available related to biomass burning in Siberia.

[Figure]

This is a unique and valuable dataset and hopefully will be built upon by this group and others, using this method in other locations and with other cores. The manuscript is well written and organized, concise, and supported by good supporting figures and supplementary data. The analytical method is adequate to make the measurements and was thoroughly tested for this particular application. The authors are cautious in the interpretation of data close to the LOD, and also apply a conservative estimate of the LOD (based on blanks rather than instrumental sensitivity). Appropriate statistical analyses are applied to the datasets produced. The authors are also careful to present the various possibilities that could impact the measured ice core acid quantities, such as surface melting, revolatilization, or atmospheric oxidation, and couch their conclusions as the likely scenarios at play, but also suggest additional studies necessary to produce a quantitative (rather than qualitative) analysis of the available data. The study is certainly of interest to the journal readership, the conclusions are supported by the available data, and the study adds important new insights to the historical biomass burning literature. I recommend publication of the manuscript in its current form.

---

## Referee Comment (RC2) · Anonymous Referee #2 · 9 Jan 2017

General comments:

Grieman et al. present a biomass burning record that is "modulated by emissions and transport" yet that provides insight into past fire activity from Siberia, a region which currently has scarce paleoclimate records published in international journals. This paper compares their results to ice cores from the perimeter of Siberia (the Altai, Kamchatka, and Greenland) to place their results into a broader perspective and demonstrating substantial regional consistency between biomass burning records. As Siberia contains vast swaths of peat and boreal forest that sequester carbon but can emit this carbon fires, having a record of past biomass burning from this region is essential to greater understanding of the carbon cycle.

The authors both modify the core's age scale, as well changing as the previous methodology for determining vanillic acid (VA) and pata-hydroxybenzoic acid in ice cores to achieve their results. This paper convincingly argues for the reasons for updating the analytical methodology. However, the paper does not sufficiently address why they changes the depth-age scale, and as this chronology is the basis of the entire work, the authors such better explain why they made such a substantial change.

Specific comments:

Abstract: The end of the discussion section provides evidence that "the similarity in timing between the Siberian biomass burning pulses, the Bond events, and the monsoonal changes likely suggests a link in this region between fires and large-scale climate variability on millennial time scales" (page 11, lines 11-13). This combination of factors is the main strength of the paper, yet the abstract only mentions the similarity between the timing of Bond evens with the timing of increased biomass burning in the Akademii Nauk ice core.

Page 1, Lines 19-21: A more recent paper, Marlon et al., 2016, demonstrates a recent rise in Northern Hemisphere biomass burning after ~2000 AD. This compilation in the Marlon et al., 2016 paper is for the entire Northern Hemisphere, verus the synthesis of high latitude Northern Hemisphere charcoal records in Marlon et al., 2008. However, the recent rise in fire activity is driven in part by newly incorporated high latitude records (such as those in Quebec) and the findings from this newer paper should be mentioned at this point. (Marlon J.R., R. Kelly, A-L. Daniau, B. Vannière, M.J. Power, P. Bartlein, P. Higuera, O. Blarquez, S. Brewer, T. Brücher, A. Feurdean, G. Gil Romera, V. Iglesias, S.Y. Maezumi, B. Magi, C.J.C. Mustaphi, and T. Zhihai. "Reconstructions of biomass burning from sediment-charcoal records to improve data-model comparisons." Biogeosciences 13 (2016): 3225-3244. DOI: 10.5194/bg-13-3225-2016)

Page 2, Lines 8 and 9: Cite Rhodes et al., 2016 (Rhodes, R. H., X. Faïn, E. J. Brook, J. R. McConnell, M. Sigl, O. Maselli, J. Edwards, C. Buizert, T. Blunier, J. Chappellaz,

J. Freitag, 2016. Local artifacts in ice core methane records caused by layered bubble trapping and in-situ production: a multi-site investigation. Climate of the Past, 12, 1061-1077, doi:10.5194/cp-12-1061-2016. http://www.clim-past.net/12/1061/2016/) regarding in situ production of methane in ice cores.

Section 2.1: As you developed an entirely new age scale for this ice core, it is essential to mention in this section why you developed this new age scale and any strengths/weaknesses of this age scale compared to the previously existing depth-age relationship.

Section 3.1: While I completely appreciate the amount of time and effort tit takes to run 3,294 samples, why were ∼700 fewer samples run for p-HBA than for VA? Were fewer samples run for p-HBA as the concentrations were often (76% of the time) below detection limits? The discussion on syringic acid in this section was one of the most interesting parts of the paper, and thank you for including this discussion the vast majority of the samples were below the detection limit, as this discussion is a service to the scientific community.

Page 7, Lines 33-34: Refer the reader back to the supplementary information regarding the age-depth scale.

Page 8, Lines 8-13: Cite these ecofloristic subdivisions in Table 1.

Page 8, Lines 13-14: Why did you start the back-trajectories at 100 m above ground level?

Page 9, Lines 14-24: To what do you ascribe the difference between the Akademii Nauk and Kamchatka Peninsula VA nad p-HBA concentrations in the 20th century? Figure 7 highlights this offset and readers are left wondering if this difference may reflect local or regional difference in biomass burning.

Page 10, Line 11-17: While describing all of the possibility interactions between the PDO and Indian Ocean monsoon is beyond the scope of this paper, mentioning that

these two phenomenon interact and are not completely separate from one another demonstrates that the cited papers are not in conflict with one another.

Conclusions: The final statement of the discussion that "the similarity in timing between the Siberian biomass burning pulses, the Bond events, and the monsoonal changes likely suggests a link in this region between fires and large-scale climate variability on millennial time scales" is the true conclusion of the paper. While the points outlined within the conclusion section are valuable, they are weaker than the finding of the connections between North Atlantic and Central Asian climate resulting in increased fire activity and then recorded in a Siberian ice core. I suggest rewriting the conclusions to emphasize the climatic aspects of this work rather than concentrating (as in the current form) on the use of VA and p-HBA as biomass burning proxies.

Figure 8: Why do you use the Pages 2K temperature reconstruction if this record does not encompass the entire time period that you are examining? If you are absolutely convinced that this temperature reconstruction must be used, then also incorporate other high latitude Northern Hemisphere temperature reconstructions that extend back through the entire record.

Technical corrections:

Abstract, line 13: Define "it" (ie. "this sudy", "or results", etc.)

Page 2, line 3: Replace "difference" with "different" and use another adjective rather than repeating "very different" twice in the same short sentence.

Page 9, Line 7: Define "this" (ie. "This result").

Page 11 Lines 11-14. You repeat the same sentence twice.

---

## Referee Comment (RC3) · Anonymous Referee #3 · 13 Jan 2017

General comments:

The manuscript by Grieman et al. presents a 3000 year record of concentration of vanillic acid and para-hydroxybenzoic acid in an ice core from the Eurasian Arctic. These two aromatic acids derive from combustion of lignin and are considered tracers of biomass burning. In the past, several authors have measured their concentrations in ice cores (McConnell et al., 2007; Kawamura et al., 2012; Müller-Tautges et al., 2016), as acknowledged by Grieman et al. However, Grieman et al. provide longer records, characterised by very high resolution, and measured with a different technique (Ion Chromatogaphy-Mass Spectrometry) than the most commonly used ones (High Performance Liquid Chromatography-MS or Gas Chromatography-MS). For these reasons,

I think the manuscript represents a substantial contribution to scientific progress. It is certainly within the scope of Climate of the Past, since biomass burning emissions contribute significantly to land carbon emissions, and fire has an important influence on ecosystems dynamics. The scientific approach and the methods are valid and the results are properly discussed. Therefore, I reccomend the manuscript for publication. However, there are a few points that need revision/explanation, including the ice dating and the back-trajectories analysis. I would like the authors to answer the following points, before the manuscript is published.

Specific comments:

Page 1, line 7: Could you provide the concentrations in [ng/l] as well (e.g.: 1 ppb=...ng/l)? I feel like [ng/l] is more widely used.

Page 1, line 10: "The timing of these periods coincides with the episodic pulsing of ice-rafted debris in the North Atlantic known as Bond events.". I would add "suggesting a link between fires and large-scale climate variability on millennial time scales"

Page 2, line 25-26: I suggest you delete "because ammonium could be derived from these other sources", as you have already explained it a few lines above.

Page 2, line 27: I would add "... while it can also originate from fossil fuel combustion during industrial times".

Page 3, line 11: You could also mention dehydroabietic acid when citing burning of conifer.

Page 3, line 16-17: In the Introduction, you have discussed the effects of emissions, transport and transformations, but I feel that you should discuss depositional and post-depositional processes more in details. What do we know about possible post-depositional processes?

Page 3 line 27: I suggest you call them "back-trajectories".

Page 4, line 24-25: Are there blanks to test any possible contamination from the melter?

Page 4, line 28-30: Why did you decide to opt for this second ice age scale? Do you consider it more accurate? If so, why? How older do you mean when you say "substantially older"? Please, provide numbers.

Page 6, line 4: A subset of how many samples were analysed using HPLC-ESI/MS/MS?

Page 6, line 26: "We are not aware of any laboratory combustion studies of the larch typically comprising the likely source regions". This sentence is not clear to me. What do you exactly mean?

Page 6, line 29: From figure 3, I would not say that the levels of VA are generally higher than those of p-HBA in the PILH. Could you specify how you have compared the levels of the two molecules (average, maxima, ...) and possibly provide the numbers for them?

Page 7, line 3-4: How did you quantify the percentage of melt layer in figure S4? Is this already published data?

Page 7, line 7: I would like you to give a definition of the LOESS smoothing either in the Methods section or here.

Page 7, line 14: I would write the periods here

Page 7, line 23: Together with the period 180-220 CE?

Page 7, line 30: If recent VA and p-HBA observations in Arctic snow and atmosphere were available, you could go one step further and use first-order assumptions on transport, transformations and post-depositional processes to estimate the atmospheric aerosol concentration of VA and p-HBA in the region of the Nauk ice core and in the source region, similarly to what done by Fischer et al. (2015) for ammonium. I don't

know whether recent VA and p-HBA observations are available, but you might want to cite Fischer et al. (2015) anyway, saying that it provides a method to estimate atmospheric concentrations in the source regions, provided that recent observations are available.

Page 8, line 12-16: There is an assumption here that you should discuss: atmospheric circulation has not changed over the past 3000 years. How likely is that assumption to be valid? Is there no evidence of changes of atmospheric circulation over the last millennia, especially over climatically relevant periods, such as the Medieval Climate Anomaly and the Little Ice Age? Some discussion is needed.

Page 10, line 4: I guess you do not mention anthropogenic influence on biomass burning because the human presence in Siberia is negligible. If so, I would spell it out, stating that the lack of significant anthropogenic activity allows you to interpret variations in biomass burning proxies and climate with a cause-effect relationship.

Page 10, line 5-7: See also Seki et al. (2015), Scientific Report

Page 10: line 32: I woul add "Figure 8, bottom plot"

Supplementary Figure S1: What do red lines/letters mean? Do they refer to the typical fragmentation in the Mass Spectrometer, as explained at page 5, line 22 of the main text? I would add some explanation in the figure's caption.

Technical comments:

Page 2, line 3: replace "difference" with "different"

Page 6, line 23: replace "gasses" with "grasses"

Page 9, line 19: there is a comma missing between p-HBA and VA: "In that study, elevated p-HBA VA,...."

―――――――――――――――

---

## Author Comment (AC2) · 24 Feb 2017

The referee raised several good points and we appreciate the comments. The manuscript has been modified as described below to take them into account. Referee comments are in italics and our responses are in a normal font.

*The end of the discussion section provides evidence that "the similarity in timing between the Siberian biomass burning pulses, the Bond events, and the monsoonal changes likely suggests a link in this region between fires and large-scale climate variability on millennial time scales" (page 11, lines 11-13). This combination of factors is the main strength of the paper, yet the abstract only mentions the similarity between the timing of Bond events with the timing of increased biomass burning in the Akademii*

*Nauk ice core.*

This point has been added to the abstract:

"...The timing of these periods coincides with the episodic pulsing of ice-rafted debris in the North Atlantic known as Bond events and a weakened Asian monsoon, suggesting a link between fires and large-scale climate variability on millennial time scales."

*Page 1, Lines 19-21: A more recent paper, Marlon et al., 2016, demonstrates a recent rise in Northern Hemisphere biomass burning after 2000 AD. This compilation in the Marlon et al., 2016 paper is for the entire Northern Hemisphere, verus the synthesis of high latitude Northern Hemisphere charcoal records in Marlon et al., 2008. However, the recent rise in fire activity is driven in part by newly incorporated high latitude records (such as those in Quebec) and the findings from this newer paper should be mentioned at this point. (Marlon J.R., R. Kelly, A-L. Daniau, B. Vannière, M.J. Power, P. Bartlein, P. Higuera, O. Blarquez, S. Brewer, T. Brücher, A. Feurdean, G. Gil Romera, V. Iglesias, S.Y. Maezumi, B. Magi, C.J.C. Mustaphi, and T. Zhihai. "Reconstructions of biomass burning from sediment-charcoal records to improve data-model comparisons." Biogeosciences 13 (2016): 3225-3244. DOI: 10.5194/bg-13-3225-2016)*

The newer Marlon et al. paper is now cited in the Introduction:

"...A synthesis of high latitude Northern Hemisphere charcoal records indicates a gradual decline in burning related to Late Holocene cooling, followed by an increase from 1750-1870 CE and a decline after 1870 CE associated with anthropogenic activity (Marlon et al., 2008). Biomass burning increased during the first half of the 20th century, declined during the second half of the 20th century, and rose sharply after 2000 CE (Marlon et al., 2016). Siberia is the largest forested area in the Northern hemisphere and Siberian wildfires constituted 5 to 20% of global biomass burning carbon emissions from 1998-2002 CE (Soja et al., 2004)... on centennial or millennial time scales with confidence (Marlon et al., 2008, 2016; Power et al., 2008)."

*Page 2, Lines 8 and 9: Cite Rhodes et al., 2016 (Rhodes, R. H., X. Faïn, E. J. Brook, J. R. McConnell, M. Sigl, O. Maselli, J. Edwards, C. Buizert, T. Blunier, J. Chappellaz,J. Freitag, 2016. Local artifacts in ice core methane records caused by layered bubble trapping and in-situ production: a multi-site investigation. Climate of the Past, 12, 1061-1077, doi:10.5194/cp-12-1061-2016. http://www.clim-past.net/12/1061/2016/) regarding in situ production of methane in ice cores.*

The Rhodes et al. (2016) paper does not appear to be relevant to the text. This paragraph is about the use of methane isotopes to constrain paleo biomass burning emissions. The local artifacts identified by Rhodes et al. (2016) would not affect the interpretation or major conclusions of the earlier studies cited here. No such claims are made in their paper. Consequently, no change was made to the manuscript.

*Section 2.1: As you developed an entirely new age scale for this ice core, it is essential to mention in this section why you developed this new age scale and any strengths/weaknesses of this age scale compared to the previously existing depth-age relationship.*

The preliminary age scale for the lower part of the ice core (Fritzsche et al., 2010) was originally based on geophysical flow modeling, but unconstrained by chemical correlations. High resolution, continuous flow analysis of multiple elements provided improved dating via identification of volcanic peaks and correlation to other ice cores.

The manuscript was changed as follows:

"The original chronology for the upper 411 m of the core (900-1998 CE) was developed based on annual layer counting of stable water isotopes and volcanic sulfate signals (Opel et al., 2013). A preliminary chronology for the lower part of the ice core (i.e. below 411m) was developed based on an adjusted Nye geophysical flow model (Fritzsche et al., 2010) but unconstrained by chemical correlations. Therefore, an alternative age model was developed and used for this study (Fig. S2). This age model is based on correlation between high resolution multi-element continuous flow measurements of

the Akademii Nauk ice core and other Arctic ice cores (Sigl et al., 2013). This new age scale yields older ages for the lower part of the ice core (below 411 m). The new age scale yields an age of 1100 BCE at 661 m as compared to 694 m for the preliminary age scale."

*why were 700 fewer samples run for p-HBA than for VA?*

Revision, page 6, line 4: "The instrument was originally optimized to analyze VA. Several samples were analyzed for VA before a method was developed analyze p-HBA."

*Page 7, Lines 33-34: Refer the reader back to the supplementary information regarding the age-depth scale.*

"The age-depth relationship for Akademii Nauk indicates constant average accumulation rate from 1700-1999 CE (140-0 m depth; Fig. S2)."

*Page 8, Lines 8-13: Cite these ecofloristic subdivisions in Table 1.*

Revised Table 1 caption:

"Table 1. Fractions of air mass back trajectories originating from or intersecting various ecofloristic zones and geographic regions (% rounded to nearest integer). Ecofloristic zones are defined by Food and Agriculture Organization (Fig. S7; http://cdiac.ornl.gov/epubs/ndp/global_carbon/carbon_documentation.html; Ruesch and Gibbs, 2008)."

*Page 8, Lines 13-14: Why did you start the back-trajectories at 100 m above ground level?*

We were curious about the influence of shallow atmospheric boundary layers or topographically influence flow on the results of the trajectories. To investigate whether such effect influenced the trajectories, we ran the trajectories at several heights from the surface to 1000 m above the surface. There was little difference in the results. Given that aerosols are deposited by both wet and dry deposition, we felt it reasonable to use

a height above surface, but still within the boundary layer.

*Page 9, Lines 14-24: To what do you ascribe the difference between the Akademii Nauk and Kamchatka Peninsula VA and p-HBA concentrations in the 20th century? Figure 7 highlights this offset and readers are left wondering if this difference may reflect local or regional difference in biomass burning.*

The increase in Kamchatka Peninsula VA and p-HBA in the 20th century is after 1970, except for a peak at 1949. There were no Akademii Nauk samples available for analysis from 1975-1983 or after 1988. It is therefore not possible to determine if the two ice core records are different in the late 20th century.

The text was revised to read: "VA and p-HBA remain elevated late in the 20th century in the ice core from the Kamchatka Peninsula. It is not possible to determine if Akademii Nauk VA and p-HBA also increase during this period due to limited Akademii Nauk sample availability after 1970."

*Page 10, Line 11-17: While describing all of the possible interactions between the PDO and Indian Ocean monsoon is beyond the scope of this paper, mentioning that these two phenomenon interact and are not completely separate from one another demonstrates that the cited papers are not in conflict with one another.*

Revision: "...drought conditions. They link Asian drought to monsoon failures during the 16th and 17th centuries. This variability may also be related to the PDO, given that the PDO can modulate the summer monsoon (Chen et al., 2013; Krishnamurthy and Krishnamurthy, 2014)."

Citations added to the manuscript: Chen,W., Feng, J., and Wu, R.: Roles of ENSO and PDO in the Link of the East Asian Winter Monsoon to the following Summer Monsoon, Journal of Climate, 26, 622–635, doi:10.1175/JCLI-D-12-00021.1, 2013.

Krishnamurthy, L. and Krishnamurthy, V.: Influence of PDO on South Asian summer monsoon and monsoon–ENSO relation, Climate Dynamics, 42, 2397–2410,

doi:10.1007/s00382-013-1856-z, 2014.

*While the points outlined within the conclusion section are valuable, they are weaker than the finding of the connections between North Atlantic and Central Asian climate resulting in increased fire activity and then recorded in a Siberian ice core. I suggest rewriting the conclusions to emphasize the climatic aspects of this work rather than concentrating (as in the current form) on the use of VA and p-HBA as biomass burning proxies.*

Revision, page 11, line 26: "Regional millennial-scale Siberian wildfire activity is not well-established due to a paucity of proxy records in the region. Siberian biomass burning may be linked to North Atlantic climate variability and the Asian monsoon. Regional records of Siberian precipitation changes would help to uncover how Bond Events may have affected climate in Siberia. More and longer records of the Arctic Oscillation and PDO are needed to reveal a relationship between Siberian biomass burning and atmospheric circulation. This study demonstrates that ice core records of organic compounds that are uniquely derived from biomass burning, such as aromatic acids, have the potential to add to our understanding of regional-scale trends in biomass burning and their relationship to climate."

*Figure 8: Why do you use the Pages 2K temperature reconstruction if this record does not encompass the entire time period that you are examining? If you are absolutely convinced that this temperature reconstruction must be used, then also incorporate other high latitude Northern Hemisphere temperature reconstructions that extend back through the entire record.*

Revision: The Marcott et al. (2013) reconstruction will also be included in Figure 8.

Revised caption: "30-year medians of domain areas of PAGES 2k temperature reconstructions (blue; PAGES 2k Consortium, 2013) and 20-year means of zonal 30-90°N stacked temperature reconstruction (black; Marcott et al., 2013),"

Added citation in manuscript, page 10, line 18: Climate reconstructions based on Northern Hemisphere proxy records show a long-term cooling trend over the past 2,000 years (Fig. 8; PAGES 2k Consortium, 2013; Hegerl et al., 2006; Ljungqvist et al., 2010; Mann et al., 2008; Marcott et al., 2013; Moberg et al., 2005)

*Abstract, line 13: Define "it" (ie. "this sudy", "or results", etc.)*

Revision: "This study clearly demonstrates that coherent aromatic acid signals are recorded in polar ice cores that can be used as proxies for past trends in biomass burning."

*Page 2, line 3: Replace "difference" with "different" and use another adjective rather than repeating "very different" twice in the same short sentence.*

Revision: "These records cover very different age ranges at varied temporal resolutions."

*Page 9, Line 7: Define "this" (ie. "This result").*

Revision: "The Akademii Nauk aromatic acid record suggests that high biomass burning emissions were sustained for multi-century periods during the last 3,000 years of the Holocene. This result perhaps indicates that the fires were widespread, but of relatively low intensity, consistent with the fact that low intensity ground fires are the principle mode of burning in Eurasian boreal forests today."

*Page 11 Lines 11-14. You repeat the same sentence twice.*

The extra sentence has been deleted.

---

## Author Comment (AC4) · 24 Feb 2017

The referee raised a few points regarding the manuscript. These points are well taken and we appreciate the comments. The manuscript has been modified to take them into account. The Referee comment is in italics and our response is in a normal font.

*Page 2, Line 20-24: Please refer that this point has been extensively addressed in Legrand et al. (2006) and it was concluded that "for Greenland ice, ammonium, formate, OC (DOC or TOC), BC, as well as vanillic and glycolic acids were enhanced well above their background values during fire events."*

[revised manuscript text omitted]

---

## Author Comment (AC5) · 26 Feb 2017

We appreciate the comments. No changes were made.

---

## Author Response (AR1)

Dear Editor,

Thank you for your comments. Below are our responses to the three issues you raised. These issues were: 1) explaining the construction of the age scale, 2) an incorrect unit conversion in the abstract, and 3) colors on Figure 8.

Your questions regarding how the time scale was constructed caused us to re-examine and revise our approach to the dating of the lowermost portion of the ice core. The revised age scale is more simply constructed and easier to explain to the reader. The time scale consists of 1) volcanic tie points correlated to other Arctic ice cores, and 2) linear extrapolation of annual layer thickness below the deepest tie point. The supplement has been modified to include a table of the tie point depths and a revised figure showing the age-depth curve. The text regarding the age scale was modified in the manuscript as shown below (section 2.1). This change to the age scale has slightly altered the interpretation of the record before 743 CE. The earliest major peak in the Akademii Nauk record and the Bond Event centered at 850 BCE no longer have similar timing. This change is reflected in the abstract, figures, and sections 3.2 and 4.3 of the manuscript. We feel confident that the major conclusions are still valid despite the changed chronology. We appreciate the attention of your comments regarding the time scale and feel that the paper is improved as a result.

"The original chronology for the upper 411 m of the core (900-1998 CE) was developed based on annual layer counting of stable water isotopes, a 1963 CE cesium peak, and 5 volcanic sulphate signals tied to Greenland ice cores (Opel et al., 2013). In this study, we used a new chronology based on continuous flow chemistry. For the upper 407 m (743-1998 CE), we interpolated linearly between a pollution signal in 1955 CE and 9 volcanic sulphur signals (Table S1; Fig. S2; Arienzo et al., 2016; Sigl et al., 2013, 2015). The dating of these tie points was based on correlation with other Arctic ice cores (Sigl et al., 2013, 2015). The original and new age scales are very similar from the surface to the depth of the Laki eruption in 1783 CE at 105 m. Below this depth, the new age scale yields progressively older ages due to the availability of volcanic tie points at 1330, 1477, 1594, and 743 CE linked to Greenland ice cores (Sigl et al., 2013, 2015). The depth-age scale below 407 m is not constrained by tie points. This part of the age scale was based on linear interpolation of annual layer thickness between the deepest tie point and the bottom of the ice core (0.21 and 0.12 m water equivalent year$^{-1}$, respectively). Layer thickness near the bed was based on oxygen isotope ($\delta^{18}$O), deuterium ($\delta$D), deuterium excess ($\delta$D-8$\delta^{18}$O), and electrical conductivity (DEP) measurements (Fritzsche et al., 2005). The lowermost portion of this age scale below the tie points is considered provisional. Further analysis of the chemistry of the lowermost portion of the ice core or other dating approaches may result in improvement in the age scale."

1. The unit conversion has been changed in the abstract to 1 ppb = 1,000 ng/l.

2. The colors for the lowermost subplots in figure 8 were exchanged.

Thank you again for your comments and we look forward to hearing from you.

Sincerely,

Mackenzie Grieman, on behalf of all coauthors

**References**

[revised manuscript text omitted]